# Neural SDEs as a Unified Approach to Continuous-Domain Sequence Modeling

## Abstract

Inspired by the ubiquitous use of differential equations to model continuous dynamics across diverse scientific and engineering domains, we propose a novel and intuitive approach to continuous sequence modeling. Our method interprets time-series data as *discrete samples from an underlying continuous dynamical system*, and models its time evolution using Neural Stochastic Differential Equation (Neural SDE), where both the flow (drift) and diffusion terms are parameterized by neural networks. We derive a principled maximum likelihood objective and a *simulation-free* scheme for efficient training of our Neural SDE model. We demonstrate the versatility of our approach through experiments on sequence modeling tasks across both embodied and generative AI [1]. Notably, to the best of our knowledge, this is the first work to show that SDE-based continuous-time modeling also excels in such complex scenarios, and we hope that our work opens up new avenues for research of SDE models in high-dimensional and temporally intricate domains.

## 1 Introduction

Sequence modeling is a fundamental task in artificial intelligence, underpinning a wide range of applications from natural language processing to time-series analysis [Sutskever, 2014, Graves and Graves, 2012, Graves, 2013]. In recent years, advances in sequence modeling have led to breakthroughs across multiple domains. Models like Generative Pre-trained Transformers (GPT) [Radford et al., 2019, Brown, 2020] have revolutionized language generation, while diffusion models have achieved state-of-the-art results in areas such as image and video generation [Ho et al., 2020, Song and Ermon, 2019, Song et al., 2021, Ho et al., 2022]. These successes underscore the importance of effective sequence modeling techniques in the development of advanced AI systems, both for discrete and continuous data.

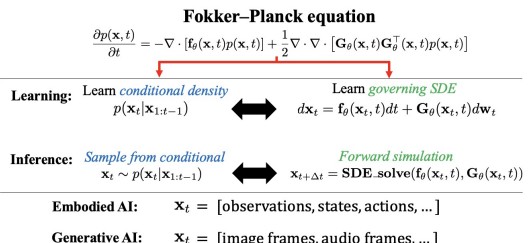

Figure 1: **A new paradigm for continuous-domain sequence modeling using SDEs.** Rather than modeling conditional densities directly, our approach represents dynamics using SDEs. The Fokker-Planck equation provides the theoretical link by describing the evolution of the probability density. This framework unifies embodied and generative AI under the same modeling paradigm.

Auto-regressive models have been the dominant approach for sequence modeling of discrete variables. This method has proven highly effective for tasks like language generation, where data is naturally discrete and sequential. However, extending auto-regressive models to continuous data is less straightforward. Continuous data often represent

---

[1]Codes and demos are accessible from our project website: `https://neuralsde.github.io/`.

smooth, time-evolving dynamics that are not naturally segmented into discrete tokens. Tokenizing continuous data can introduce quantization errors and obscure important information such as the closeness between states induced by the distance metric of the continuous state space [Argall et al., 2009, Li et al., 2024]. To model such data effectively, it's essential to preserve their inherent continuity and stochasticity.

Recent methods like diffusion models [Ho et al., 2020, Song et al., 2021] and flow matching frameworks [Lipman et al., 2022] have made significant progress in handling continuous variables by leveraging stochastic processes. These approaches transform a simple initial distribution into a complex target distribution through a series of learned transformations, effectively capturing the data distribution in continuous spaces. However, in the context of sequence modeling, this iterative approach can be less natural and computationally intensive, which often involves a high transport cost and requires many iterative steps to produce satisfactory results [Tong et al., 2023, Kornilov et al., 2024]. In contrast, a more intuitive and efficient strategy is to model the sequence by directly learning the transitions between consecutive states, reflecting the inherent temporal continuity of the data.

These limitations highlight the need for sequence models designed specifically for continuous data, capable of capturing both the deterministic trends and stochastic fluctuations of the underlying dynamics without relying on extensive iterative transformations.

**Differential Equations as a Unifying Framework.** In many real-world applications, continuous dynamical systems are naturally described by ordinary or stochastic differential equations. From physical processes like fluid flow and orbital mechanics [Gianfelici, 2008, Harier et al., 2000] to robotic control and autonomous systems [Zhu, 2023], the underlying dynamics are often given—or well approximated—by state-space differential equations that evolve continuously in time.

By leveraging Neural Stochastic Differential Equations (Neural SDEs) [Kong et al., 2020, Kidger et al., 2021a], we propose a method that models the time evolution of continuous systems directly. This approach maintains the intrinsic continuity of the data and provides a more natural representation of continuous-time processes compared to traditional discrete-time or tokenization-based methods. In summary, the contributions of this paper include:

- A novel Neural SDE framework for continuous sequence modeling utilizing a simulation-free maximum-likelihood method with a decoupled two-stage optimizer, eliminating the costly forward simulations required in traditional gradient estimation.

- Significant improvements in inference efficiency due to directly modeling continuous-time dynamics—rather than discretely transitioning from Gaussian noise as in diffusion and flow matching approaches—leading to substantially fewer function evaluations (NFEs).

- Empirical validations demonstrating our method's unique advantages, including modeling of multi-modal distributions, robustness to irregular dynamics and natural temporal interpolation, while achieving performance comparable to baselines across diverse tasks such as imitation learning and video prediction.

## 2 Backgrounds

In this section, we provide an overview of foundational concepts relevant to our approach. We begin with diffusion models and their extension to continuous stochastic differential equations (SDEs). We then discuss the flow matching and stochastic interpolant frameworks, which are instrumental in formulating our method. Finally, we introduce Neural SDEs and highlight the challenges associated with their training.

### 2.1 Diffusion Models and Their SDE Formulation

Diffusion models have become a cornerstone in generative modeling, achieving remarkable success in generating high-fidelity images and other data types [Ho et al., 2020, Song et al., 2021]. At their core, these models involve a forward process that gradually adds noise to the data and a reverse process that reconstructs the data by denoising.

### 2.1.1 Discrete Diffusion Models

In the discrete setting, the forward diffusion process is defined over discrete time steps, where Gaussian noise is incrementally added to the data. The reverse process involves learning to denoise the corrupted data to recover the original data distribution. This is typically achieved by training a neural network to predict the noise added at each step, using a mean squared error loss between the predicted and true noise.

### 2.1.2 Continuous SDE Formulation

The continuous-time formulation of diffusion models represents the forward and reverse processes as solutions to SDEs.

The forward diffusion process is defined by the SDE:

$$d\mathbf{x}_t = \mathbf{f}(\mathbf{x}_t, t)dt + g(t)d\mathbf{w}_t, \tag{1}$$

where $\mathbf{x}_t \in \mathbb{R}^d$ is the data at time $t \in [0, T]$, $\mathbf{f}(\mathbf{x}_t, t)$ is the flow coefficient, $g(t)$ is the diffusion coefficient, and $\mathbf{w}_t$ is a standard Wiener process.

In the case of the Variance Preserving (VP) SDE [Song et al., 2021], which corresponds to the discrete diffusion model with a variance schedule $\beta_t$, the flow and diffusion coefficients are predefined functions given by:

$$\mathbf{f}(\mathbf{x}_t, t) = -\frac{1}{2}\beta(t)\mathbf{x}_t, \ g(t) = \sqrt{\beta(t)}. \tag{2}$$

Here, $\beta(t)$ is a continuous-time version of the noise schedule from the discrete model.

### 2.1.3 Reverse SDE and Score Function

To generate data samples, we consider the reverse-time SDE that has the same marginal distribution as the forward SDE:

$$d\mathbf{x}_t = \left[\mathbf{f}(\mathbf{x}_t, t) - g^2(t)\nabla_{\mathbf{x}_t} \log q_t(\mathbf{x}_t)\right] dt + g(t)d\bar{\mathbf{w}}_t, \tag{3}$$

where $q_t(\mathbf{x}_t)$ is the marginal distribution of $\mathbf{x}_t$ at time $t$, and $\bar{\mathbf{w}}_t$ is a reverse-time Wiener process. The term $\nabla_{\mathbf{x}_t} \log q_t(\mathbf{x}_t)$ is the *score function*, representing the gradient of the log probability density at time $t$.

In practice, since $q_t(\mathbf{x}_t)$ is unknown, the score function is approximated by a neural network $\mathbf{s}_\theta(\mathbf{x}_t, t)$, trained to minimize the score-matching objective which corresponds to the denoising objective in the discrete formulation:

$$\mathcal{L}_{\text{score}} = \mathbb{E}_t \left[\lambda(t)\mathbb{E}_{\mathbf{x}0} \left[\|\mathbf{s}_\theta(\mathbf{x}_t, t) - \nabla_{\mathbf{x}_t} \log q(\mathbf{x}_t \mid \mathbf{x}_0)\|^2\right]\right], \tag{4}$$

where $\mathbf{x}_0$ is the original data sample, $\mathbf{x}_t$ is obtained by solving the forward SDE starting from $\mathbf{x}_0$, $\lambda(t)$ is a weighting function, and $q(\mathbf{x}_t \mid \mathbf{x}_0)$ is the transition kernel of the forward SDE. In the case of linear Gaussian transition as in Eq. (2), $\nabla_{\mathbf{x}_t} \log q(\mathbf{x}_t \mid \mathbf{x}_0)$ has a simple closed-form solution which is proportional to the sampled noise.

## 2.2 Flow Matching and Stochastic Interpolants

Instead of learning SDEs, Flow matching and Stochastic Interpolants provide frameworks for learning a diffusion-free ordinary differential equation (ODE) for generative modeling.

### 2.2.1 Interpolative Approach

Both flow matching [Lipman et al., 2022] and stochastic interpolants [Albergo et al., 2022] rely on constructing an *interpolant* between a simple base distribution $p_0(\mathbf{x})$ and a target distribution $p_1(\mathbf{x})$. The interpolant is defined as a time-dependent process $\mathbf{x}_t$ that smoothly transitions from $\mathbf{x}_0 \sim p_0$ to $\mathbf{x}_1 \sim p_1$ as $t$ goes from 0 to 1.

An example of a stochastic interpolant is:

$$\mathbf{x}_t = \alpha(t)\mathbf{x}_0 + \beta(t)\mathbf{x}_1 + \sigma(t)\boldsymbol{\xi}, \tag{5}$$

where $\alpha(t)$ and $\beta(t)$ are deterministic scalar functions satisfying $\alpha(0) = 1$, $\alpha(1) = 0$, $\beta(0) = 0$, $\beta(1) = 1$, $\sigma(t)$ controls the magnitude of the stochastic component, and $\boldsymbol{\xi} \sim \mathcal{N}(\mathbf{0}, \mathbf{I})$ is standard Gaussian noise.

### 2.2.2 Simulation-Free Training Objective

The key idea is to learn a vector field $\mathbf{v}_\theta(\mathbf{x}_t, t)$ such that the time derivative of the interpolant matches the vector field:

$$d\mathbf{x}_t = \mathbf{v}_\theta(\mathbf{x}_t, t)dt. \tag{6}$$

The training objective minimizes the expected squared difference between the model's vector field and the true time derivative of the interpolant:

$$\mathcal{L}_{\text{FM}} = \mathbb{E}_t \left[ \mathbb{E}_{\mathbf{x}_0, \mathbf{x}_1, \boldsymbol{\xi}} \left[ \| \mathbf{v}_\theta(\mathbf{x}_t, t) - \frac{d\mathbf{x}_t}{dt} \|^2 \right] \right]. \tag{7}$$

Because $\mathbf{x}_t$ and $\frac{d\mathbf{x}_t}{dt}$ can be computed analytically from (5), training does not require simulating the dynamics of $\mathbf{x}_t$.

## 2.3 Neural Stochastic Differential Equations

Neural Stochastic Differential Equations (Neural SDEs) [Li et al., 2020a, Kidger et al., 2021b] extend Neural ODEs [Chen et al., 2018] by incorporating stochastic components into the system dynamics. A Neural SDE models the evolution of a stochastic process as:

$$d\mathbf{x}_t = \mathbf{f}_\theta(\mathbf{x}_t, t)dt + \mathbf{G}_\theta(\mathbf{x}_t, t)d\mathbf{w}_t, \tag{8}$$

where $\mathbf{f}_\theta$ and $\mathbf{G}_\theta$ are neural networks parameterized by $\theta$.

Training Neural SDEs involves computing gradients of a loss function with respect to the parameters $\theta$. A common approach is the *adjoint sensitivity method* [Li et al., 2020b], which computes these gradients by solving an adjoint SDE backward in time alongside the forward simulation. While this method is memory-efficient, it poses computational challenges:

- **Computational Overhead**: Simulating both the forward and backward SDEs increases computational cost.
- **Numerical Stability**: Backward integration through stochasticity can introduce numerical instability, as stochastic differential equations are less straightforward to reverse due to their inherent randomness.

These challenges motivate the exploration of alternative training methods that can efficiently handle Neural SDEs without the need for backward simulation through stochastic processes.

## 2.4 Connections and Motivations

The continuous SDE formulation of diffusion models and the flow matching framework provide powerful tools for modeling data distributions through stochastic processes. However, these methods often assume fixed diffusion coefficients or rely on extensive iterative computations. While Neural SDEs have been employed for continuous sequence modeling [Oh et al., 2024, Tzen and Raginsky, 2019], gradient computation under these formulations typically incurs a cost that scales linearly with the number of simulation steps. This results in significant overhead for high-dimensional data or models with large parameter counts, as most existing differentiation methods for Neural SDEs—including pathwise automatic differentiation [Tzen and Raginsky, 2019, Li et al., 2020b], adjoint sensitivity analysis[Li et al., 2020b, Blasingame and Liu, 2024], and likelihood-ratio-based estimators[Glynn, 1990]—require full forward (and sometimes backward) integration of the SDE per training sample.

Our approach leverages Neural SDEs' ability to model both drift and diffusion—faithfully capturing intrinsic data uncertainty and stochasticity—while adopting the simulation-free training paradigm pioneered by flow matching (which itself is derived from an optimal-transport control formulation, not likelihood). Crucially, we re-derive this flow regression from a maximum-likelihood (ML) objective and extend it with a decoupled two-stage optimizer: first fitting the flow via our ML-driven flow-matching analogue, then separately estimating diffusion coefficients through a residual log-likelihood objective. This two-stage scheme entirely avoids forward SDE unrolling.

By formulating the problem as learning the parameters of an SDE that describes the evolution of continuous sequences, we directly model the time dynamics without the need for discretization or high transport costs associated with iterative methods.

# 3  Our Approach

We introduce a Neural SDE framework to model continuous-time sequences, capturing both deterministic trends and stochastic fluctuations inherent in the data. Our model learns a **time-invariant** SDE of the form:

$$d\mathbf{x}_t = \mathbf{f}(\mathbf{x}_t)dt + \mathbf{g}(\mathbf{x}_t) \odot d\mathbf{w}_t, \tag{9}$$

Our goal is to learn $\mathbf{f}$ and $\mathbf{g}$ from data such that the SDE accurately represents the underlying continuous-time dynamics observed in the sequences.

Compared with the general SDE (Eq. 8), we are making two key modeling choices:

1. **Time-Invariant System:** The flow and the diffusion function are implicit function of time through the dependency on the state vector, because many real-world systems exhibit dynamics that are consistent over time [Zhu, 2023, Slotine et al., 1991]. Modeling these systems with a time-invariant SDE simplifies the learning task and captures the essential state-dependent dynamics without unnecessary complexity.

2. **Diagonal Diffusion Matrix:** For computational tractability, we use a diagonal diffusion matrix:

$$\mathbf{g}(\mathbf{x}_t) = \mathrm{diag}\left(\sigma_1(\mathbf{x}_t), \sigma_2(\mathbf{x}_t), \ldots, \sigma_d(\mathbf{x}_t)\right), \tag{10}$$

instead of a full matrix. This implies that the stochastic components affecting each state dimension are independent. The diagonal assumption simplifies calculations, particularly the inversion and determinant of the covariance matrix during training, leading to more efficient optimization. Moreover, in many practical applications, the noise affecting different state variables can be reasonably considered uncorrelated [Gardiner, 2009, Oksendal, 2013].

## 3.1  Negative Log-Likelihood Derivation

Using Euler–Maruyama discretization (detailed in Appendix A) and the resulting Gaussian transition probability (Appendix B), the negative log-likelihood (NLL) for a single transition segment simplifies to:

$$\mathcal{L}_{\mathbf{x}_t} = \frac{1}{2}\|(\mathbf{f}(\mathbf{x}_t) - \frac{\Delta \mathbf{x}}{\Delta t}) \odot \mathbf{g}(\mathbf{x}_t)^{-1}\|^2 \Delta t + \frac{1}{2}\log \det \mathbf{g}(\mathbf{x}_t)\mathbf{g}(\mathbf{x}_t)^\top$$

$$= \frac{1}{2}\sum_{i=1}^{d}\left(\frac{f_i(\mathbf{x}_t) - \frac{\Delta x_i}{\Delta t_i}}{\sigma_i(\mathbf{x}_t)}\right)^2 \Delta t_i + \frac{1}{2}\sum_{i=1}^{d}\log \sigma_i^2(\mathbf{x}_t), \tag{11}$$

where $\Delta \mathbf{x} = \mathbf{x}_{t+\Delta t} - \mathbf{x}_t$, and we have omitted constant terms that do not depend on the modeling parameters.

This loss function consists of:

- **Prediction Error Term (first):** Measures how well the flow function $\mathbf{f}(\mathbf{x}_t)$ matches the observed state changes $\Delta \mathbf{x}$. Note that this term is scaled by inverse of diffusion term $\mathbf{g}(\mathbf{x}_t)$, which is absent in Flow-Matching methods. The intuition is that a mismatch between the flow and the observed state change can be attributed to either an inaccurate flow prediction or the intrinsic uncertainty of the process measured by the diffusion term.

- **Complexity Penalty Term (second):** The logarithmic determinant term regularizes overly large diffusion coefficients, preventing the model from assigning high uncertainty indiscriminately.

Given a trajectory of observed data points $\{\mathbf{x}_{t_k}\}_{k=0}^{N}$ at times $\{t_k\}_{k=0}^{N}$, we can aggregate the single-segment losses to obtain the total loss:

$$\mathcal{L} = -\sum_{k=0}^{N-1}\log p(\mathbf{x}_{t_{k+1}} \mid \mathbf{x}_{t_k}) = \sum_{k=0}^{N-1}\mathcal{L}_{\mathbf{x}_{t_k}}. \tag{12}$$

Unlike Auto-regressive models where $\mathbf{x}_{t_{k+1}}$ needs to be conditioned on all the previous states, here in our model the conditional distribution is simply between consecutive steps due to the the markovian property of the SDE.

## 3.2 Training Strategy: Decoupled Optimization of Flow and Diffusion

The first term in Eq. (11) depends on both $\mathbf{f}$ and $\mathbf{g}$, we empirically observe that joint training is susceptible to getting stuck at local minimum. To enhance training stability and interpretability, we derive a decoupled optimization scheme for the flow $f$ and the diffusion term $\mathbf{g}$.

We notice that Eq. (11) can be analytically minimized w.r.t. $\mathbf{g}$ by setting

$$g_i^2(\mathbf{x}_t) = \sigma_i^2(\mathbf{x}_t) = \left( f_i(\mathbf{x}_t) - \frac{\Delta x_i}{\Delta t_i} \right)^2 \Delta t_i. \tag{13}$$

Plugging Eq. (13) back to Eq. (11) and discarding constant terms, we get the following simplified objective for the flow term

$$\mathcal{L}_{\mathbf{x}_t}^f = \frac{1}{2} \sum_{i=1}^d \log \left( f_i(\mathbf{x}_t) - \frac{\Delta x_i}{\Delta t_i} \right)^2. \tag{14}$$

Likewise, the constraint Eq. (13) suggests the following objective for the diffusion term

$$\mathcal{L}_{\mathbf{x}_t}^g = \frac{1}{2} \sum_{i=1}^d \left( \sigma_i^2(\mathbf{x}_t) - \left( f_i(\mathbf{x}_t) - \frac{\Delta x_i}{\Delta t_i} \right)^2 \Delta t_i \right)^2. \tag{15}$$

This objective suggests that the diffusion coefficients are matching the residual error of the flow prediction. Similarly, the training objective for the whole trajectory can be obtained by aggregating all the segments as in Eq. (12).

## 3.3 Implications of the Simplified Flow Objective

The logarithmic-squared flow loss (Eq. 14) provides two primary benefits: **(i)** *scale invariance* across different dimensions, which eliminates the need for manually tuning per-dimension loss weights—a property particularly valuable in multi-modal settings where each modality may have distinct units or dynamic ranges; and **(ii)** *robustness to large errors* due to sub-linear growth, allowing the model to handle high-variance data by naturally increasing the learned diffusion term rather than incurring large penalties. A detailed analysis and discussion appear in Appendix C.

## 3.4 Implementation Details

Our main derivation thus far focuses on the theoretical formulation. In practice, we make several additional choices to improve training stability and performance. For example, we employ *data interpolation* between observed time steps, *noise injection* for regularization, and an optional *denoiser* network for improving inference-time trajectory likelihood. We also discuss how we handle the numerical stability of the logarithmic loss and datasets without explicit time. Full details about these implementation aspects are provided in Appendix D.

# 4 Experiments

In this section, we empirically evaluate the capabilities of Neural SDE across three distinct sequence modeling tasks: (1) a 2D Bifurcation task designed to assess multi-modal trajectory generation, (2) the Push-T imitation learning task, and (3) video prediction on standard benchmark datasets. Details about the setup are provided in the Appendix E.

## 4.1 2D Branching Trajectories

Accurately capturing multi-modal distributions is crucial for sequence modeling tasks. To evaluate the ability of our model to capture multi-modal distributions, we designed a simple 2D trajectory generation task, where the ground truth trajectories exhibit a Y-shaped bifurcation pattern.

In Appendix E.1, Figure 4 shows the generated trajectories for our approach and diffusion/flow-matching approaches at both low and high densities (number of steps per trajectory). At the low density, all three models successfully capture the bimodal distribution, producing trajectories that

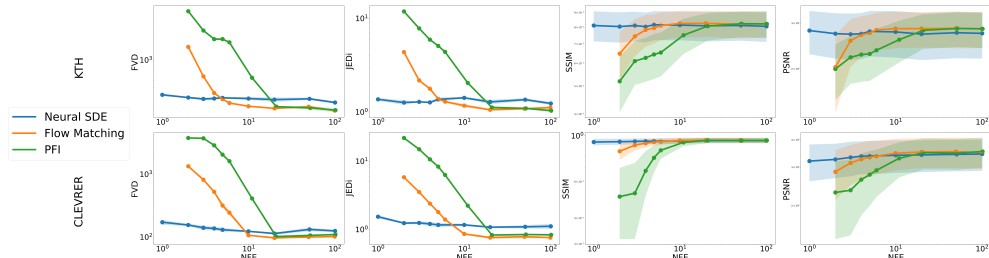

Figure 2: **Inference Efficiency**. The plots show the performance of Neural SDE, Flow Matching, and PFI on the KTH and CLEVRER datasets, measured by the metrics FVD, JEDI, SSIM, and PSNR, with respect to the number of function evaluations (NFE). Lower FVD and JEDI and higher SSIM and PSNR indicate better performance. To control the NFEs of PFI and Neural SDE, we use fixed step sizes. All metrics are estimated by sampling 256 test video sequences (with replacement), and the evaluation procedure is repeated 4 times. We report the mean performance across the 4 runs, along with the 95% confidence interval bands.

branch as expected. However, a significant difference occurs at the high density. While Neural SDE still captures the two branches, DDIM and Rectified Flow fail. We speculate that this degradation in performance for DDIM and Rectified Flow at higher density is due to covariate-shift.

To verify, we show the contribution of different components within our model in Figure 5. Comparing with the *Flow* and *Flow+Diffusion* plots, we see that the addition of the diffusion term introduces stochasticity into the trajectories, leading to branching. The *Flow+Denoiser+Diffusion* plot shows that incorporating the denoiser effectively mitigates covariate-shift.

## 4.2 Push-T

To evaluate the effectiveness of our Neural SDE in imitation learning, we employ the Push-T task. Details about the setup are provided in appendix E.2.

**Results** Table 1 shows the performance of our Neural SDE compared to the Diffusion Policy baseline. Our model achieves competitive performance, demonstrating its effective-

| Method | LSTM-GMM | IBC | BET | DP | NSDE |
|---|---|---|---|---|---|
| TAC↑ | 0.67 | 0.90 | 0.79 | 0.95 | 0.92 |

Table 1: **Target area coverage (TAC) comparison on PushT.** Baseline results (trained on a Transformer backbone) are from Chi et al. [2023]. Neural SDE uses an MLP composed of four residual blocks, each containing two linear layers with a skip connection.

ness in policy learning. This is notable given that SDEs often assume some level of smoothness in the underlying dynamics, while imitation learning often involves non-smooth dynamics. To further investigate the ability of our model to handle trajectories with sharp transitions, we visualize a representative push-T trajectory in Figure 6. This figure shows that Neural SDE can effectively model such non-smooth behavior, demonstrating its robust sequence-modeling capability.

## 4.3 Video Prediction

To assess our model's ability to learn complex temporal dynamics, we evaluate its performance on two video prediction benchmarks: the KTH Schuldt et al. [2004] dataset and the CLEVRER [Yi et al., 2019] dataset. Details about the setup are provided in appendix E.3.

**Quantitative Comparison and Inference Efficiency** Our Neural SDE model achieves comparable results to the Flow Matching [Lipman et al., 2022] and Probabilistic Forecasting with Interpolants (PFI) [Chen et al., 2024] baselines across multiple metrics, including FVD Unterthiner et al. [2019], JEDI [Bicsi et al., 2023] , SSIM [Wang et al., 2004], and PSNR [Hore and Ziou, 2010], on the KTH and CLEVRER datasets. One of the most compelling advantages of our method is its inference efficiency. Figure 2 shows the relationship between the number of function evaluations (NFE) and performance metrics. Our approach significantly outperforms the baselines in efficiency, requiring only 2 steps to achieve results comparable to Flow Matching and PFI, which typically need 5 to 20 steps to generate future frames of reasonable quality. This substantial reduction in NFE highlights the

intrinsic efficiency of directly modeling continuous-time dynamics, making it an appealing solution to sequence modeling tasks of high computational complexity.

**Temporal Resolution and Implicit Interpolation** Our Neural SDE approach also offers an appealing capability of improving temporal resolution without incurring additional training cost. As demonstrated in Appendix F, Neural SDEs can generate coherent intermediate frames between two consecutive frames in the original dataset. In contrast, flow-based methods like Flow Matching are limited to generating frames at the specific temporal resolution defined by the training data, requiring retraining or ad hoc interpolation techniques to achieve higher frame rates. This "free" interpolation capability underscores the power of modeling video as a continuous process.

**Scalability Analysis** We further investigated the scalability of our Neural SDE model by analyzing the relationship between model size and validation loss. Following the methodology in [Kaplan et al., 2020, Tian et al., 2024], we varied the depth and hidden dimensions of the U-ViT

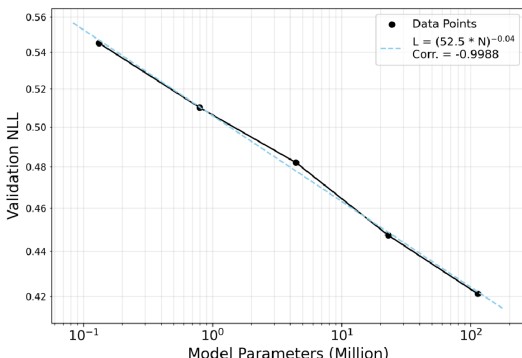

Figure 3: **Scaling law of Neural SDE with U-ViT backbone on the CLEVRER dataset.** The Pearson correlation coefficient ($-0.9988$) indicates a strong power-law relationship, suggesting that increasing model size leads to improved performance.

backbone used in our model, resulting in models ranging from $0.13$ million to $113.44$ million parameters, see Appendix H for detail.

Figure 3 shows the scaling behavior of our model. We observe a clear power-law relationship between the number of model parameters and the validation loss. This scaling behavior suggests that our Neural SDE architecture can leverage larger model capacities to achieve better video prediction performance.

## 5 Related Works

A large class of related works are *bridging-based* generative models, including diffusion approaches [Ho et al., 2020, Song et al., 2021], flow-based or flow-matching frameworks [Dinh et al., 2016, Ho et al., 2019, Kobyzev et al., 2020, Lipman et al., 2022, Hu et al., 2024], stochastic interpolants [Albergo et al., 2022, Chen et al., 2024, Albergo et al., 2023], and Schrödinger bridges [De Bortoli et al., 2021, Liu et al., 2023]. Despite varying details, these methods commonly start from a simple (often Gaussian) prior distribution and *transport* it to the target data distribution via learned transformations. This "from-noise-to-data" perspective has driven state-of-the-art results in static, unpaired data domains (e.g., image generation). However, two main issues arise when directly applying such frameworks to *continuous time-series*: **(1)** they generally unroll samples all the way from an uninformative prior, incurring large transport costs for dense or high-dimensional sequences; and **(2)** they often treat time as a "pseudo-time" schedule (e.g., noise levels) rather than the actual chronological axis of the system. As a result, bridging-based methods can be less intuitive and potentially suboptimal for tasks where the *true* temporal progression—and possibly irregular sampling—is essential.

Meanwhile, *Neural SDE* approaches [Kong et al., 2020, Li et al., 2020a, Kidger et al., 2021a, Liu et al., 2020, Park et al., 2021, Djeumou et al., 2023, Tzen and Raginsky, 2019, Oh et al., 2024] have emerged to incorporate Brownian noise into Neural ODEs, targeting time-series forecasting or sporadic data interpolation. For example, SDE-Net [Kong et al., 2020] and Neural SDE [Li et al., 2020a] inject noise into network layers for uncertainty estimation or robustness, while others develop specialized SDE solvers or variational methods to handle irregular temporal observations [Kidger et al., 2021a, Liu et al., 2020, Oh et al., 2024]. Although conceptually related, these works often employ *complex* training objectives (e.g., forward-backward SDE solves or Bayesian evidence bounds) and focus on low-dimensional *time-series* datasets rather than more complex tasks as imitation learning or video prediction.

By contrast, our *Neural SDE* framework adopts a *direct maximum-likelihood* approach for learning continuous-time dynamics from observed consecutive states. Rather than bridging data from a simple noise prior, each pair of adjacent observations is treated as a sample from an underlying SDE, bypassing the overhead of unrolling from isotropic noise. This design both **respects real time evolution**—via Flow and diffusion that capture deterministic trends and random fluctuations—and **streamlines conditional modeling**, since every transition likelihood is straightforwardly evaluated in a Markovian manner. Consequently, our method is more natural for *embodied and generative AI*, where continuous trajectories must be modeled reliably, yet classical bridging-based methods may be unnecessarily complex and less aligned with genuine temporal structure.

## 6    Limitations and Future Work

Our formulation rests on a time invariant Markov assumption: the transition kernel $p(x_{t+\Delta t}|x_t)$ is time-independent, so past influences are encoded only through the current state. It can miss long-range non-Markovian dependencies; on KTH, for instance, visually similar clips occasionally trigger unintended action switches. Moving forward, relaxing this assumption by introducing time-varying dynamics or slower-evolving latent variables could provide the model with a memory mechanism to better capture long-horizon dependencies. Second, we model diffusion coefficients with a diagonal covariance for tractability; this excludes cross-dimensional noise correlations and may under-represent uncertainty in coupled systems. Exploring richer noise models such as full or low-rank covariance structures could allow the model to represent inter-variable stochastic coupling without sacrificing efficiency. Third, the weighting coefficient $\alpha$ in the denoiser network is manually selected per dataset, adding another hyperparameter. Automating the selection of $\alpha$ either by learning it directly from data or adapting it dynamically—could eliminate manual tuning and improve generalization across tasks.

## 7    Conclusions

We have presented a novel *Neural SDE* framework for continuous-domain sequence modeling, offering an alternative to existing *bridging-based* approaches such as diffusion and flow-matching methods. By learning both drift and diffusion terms via a direct maximum likelihood objective, our method naturally captures the stochastic and deterministic components of time-evolving data. Moreover, the Markovian formulation circumvents the need for iterative unrolling from a simple noise prior, thus reducing transport costs and simplifying inference.

Through experiments on multiple domains, we demonstrated that Neural SDEs (1) faithfully model multi-modal distributions, (2) handle sharp or irregular dynamics, (3) generate high-quality predictions with few inference steps, and (4) offer "free" temporal interpolation beyond the training schedule. Our analysis also highlights key properties such as the *scale invariance* of the log-flow loss and the interpretability gained by explicitly modeling diffusion.

Looking ahead, an interesting extension would be to incorporate additional conditional variables for capturing longer histories, rather than the current approach of augmenting only the latest state. This could improve performance in tasks requiring extended temporal memory. Another promising direction involves tackling noisy actions in real-world embodied AI by focusing on state or observation transitions paired with a learned inverse-dynamics model—alleviating the need for accurately recorded actions. We hope this work stimulates further study into *Neural SDEs* as a unified, principled approach to continuous-domain sequence modeling, bridging the gap between traditional differential equation frameworks and modern machine learning techniques.

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

## A  Euler–Maruyama Discretization

To work with discrete data, we discretize the continuous SDE using the Euler–Maruyama method [Bayram et al., 2018]. Over a small time interval $\Delta t$, the discretized SDE approximates the evolution of $\mathbf{x}_t$ as:

$$\mathbf{x}_{t+\Delta t} = \mathbf{x}_t + \mathbf{f}(\mathbf{x}_t)\Delta t + \mathbf{g}(\mathbf{x}_t) \odot \Delta \mathbf{w}_t, \tag{16}$$

where $\Delta \mathbf{w}_t = \mathbf{w}_{t+\Delta t} - \mathbf{w}_t$ is the increment of the Wiener process over $\Delta t$. Since $\Delta \mathbf{w}_t \sim \mathcal{N}(\mathbf{0}, \Delta t \mathbf{I}_d)$, the stochastic term introduces Gaussian noise with covariance proportional to $\Delta t$.

# B  Transition Probability

Under this discretization, the conditional distribution of the next state $\mathbf{x}_{t+\Delta t}$ given the current state $\mathbf{x}_t$ is Gaussian:

$$p(\mathbf{x}_{t+\Delta t} \mid \mathbf{x}_t) = \mathcal{N}\left(\mathbf{x}_{t+\Delta t}; \boldsymbol{\mu}_t, \boldsymbol{\Sigma}_t\right), \tag{17}$$

where:

$$\boldsymbol{\mu}_t = \mathbf{x}_t + \mathbf{f}(\mathbf{x}_t)\Delta t, \ \boldsymbol{\Sigma}_t = \mathbf{g}(\mathbf{x}_t)\mathbf{g}(\mathbf{x}_t)^\top \Delta t. \tag{18}$$

Under the diagonal form of $\mathbf{g}(\mathbf{x}_t)$, the covariance matrix $\boldsymbol{\Sigma}_t$ simplifies to:

$$\boldsymbol{\Sigma}_t = \text{diag}\left(\sigma_1^2(\mathbf{x}_t), \sigma_2^2(\mathbf{x}_t), \ldots, \sigma_d^2(\mathbf{x}_t)\right)\Delta t. \tag{19}$$

This probabilistic description allows us to compute the likelihood of observed data under our model.

# C  Implications of the Simplified Flow Objective

The simplified flow objective in Eq. (14) introduces two key advantages:

**Scale-Invariance:**  The logarithmic squared loss imparts a *scale-invariant* property to the objective function. Specifically, scaling each dimension of the flow function $f_i(\mathbf{x}_t)$ and the observed rate $\frac{\Delta x_i}{\Delta t_i}$ by independent positive constants $c_i$ adds only a constant term to the loss:

$$\begin{aligned}
\mathcal{L}_{\mathbf{x}_t}^f &= \frac{1}{2}\sum_{i=1}^{d}\left[\log\left(c_i^2\left(f_i(\mathbf{x}_t) - \frac{\Delta x_i}{\Delta t_i}\right)^2\right)\right] \\
&= \frac{1}{2}\sum_{i=1}^{d}\left[\log c_i^2 + \log\left(f_i(\mathbf{x}_t) - \frac{\Delta x_i}{\Delta t_i}\right)^2\right].
\end{aligned} \tag{20}$$

The additive constants $\log c_i^2$ do not affect optimization with respect to model parameters. This property is particularly beneficial when dealing with multi-modal data where different dimensions have varying units or scales—as in embodied AI scenarios involving diverse physical quantities. It eliminates the need for manual weighting or scaling of loss terms across dimensions.

**Robustness to Large Errors**  The logarithmic function grows sub-linearly, making the loss function **more tolerant of large errors** compared to a standard squared loss. This aligns with the model's treatment of uncertainty via the diffusion term $\mathbf{g}(\mathbf{x}_t)$. Larger discrepancies between the predicted flow and observed changes are accommodated as increased stochasticity rather than penalized heavily. This characteristic enables the model to handle systems with inherent variability more effectively, without being unduly influenced by outliers or noise.

# D  Implementation Details

**Algorithm Overview**  Our Neural SDE learning algorithm consists of the following high-level steps:

---
**Algorithm 1** Neural SDE Learning Algorithm

---
**Require:** Dataset of transition tuples $\mathcal{D} = \{\mathbf{x}_{t_i}, \mathbf{x}_{t_{i+1}}\}_{i=1}^{N}$
**Require:** Initialized neural networks: flow $\mathbf{f}(\mathbf{x}; \theta_f)$, diffusion $\boldsymbol{\sigma}^2(\mathbf{x}; \theta_g)$, and optional denoiser $\mathbf{d}(\mathbf{x}; \theta_d)$
**Ensure:** Trained parameters $\theta_f, \theta_g, \theta_d$
 1: **for** each mini-batch transitions in $\mathcal{D}$ **do**
 2:   **Interpolate** between observed states to obtain $\mathbf{x}_\tau$
 3:   **Add noise** to interpolated states: $\tilde{\mathbf{x}}_\tau = \mathbf{x}_\tau + \boldsymbol{\eta}$
 4:   **Update flow network** parameters $\theta_f$ using the flow loss (Eq.14)
 5:   **Update diffusion network** parameters $\theta_g$ using the diffusion loss (Eq.15)
 6:   **(Optional) Update denoiser network** parameters $\theta_d$ using the denoising score matching loss (Eq.22)
 7: **end for**

---

**State Interpolation for Training**   To augment the training data and improve generalization, we interpolate between observed discrete states, similar to techniques used in flow matching and diffusion models. For each pair of consecutive observed states $\mathbf{x}_{t_k}$ and $\mathbf{x}_{t_{k+1}}$, we sample intermediate times $\tau \in [t_k, t_{k+1}]$ uniformly and generate linearly interpolated states $\mathbf{x}_\tau$. This strategy can be interpreted as using a modified discretization scheme that introduces additional evaluation points without altering the order of discretization error inherent in the Euler–Maruyama method. By incorporating interpolated states, we effectively increase the diversity of training samples and encourage the model to capture the underlying continuous dynamics more accurately.

**Noise Injection**   Inspired by the stochastic interpolants framework [Albergo et al., 2022], we add random noise to the interpolated states during training. Specifically, we perturb the states with Gaussian noise: $\tilde{\mathbf{x}}_\tau = \mathbf{x}_\tau + \boldsymbol{\eta}$, where $\boldsymbol{\eta} \sim \mathcal{N}(\mathbf{0}, \sigma^2 \mathbf{I})$. This can be seen as modifying the discretization scheme to account for stochastic variability, while maintaining the same order of discretization error. Noise injection acts as an implicit regularizer, enhancing the model's robustness to data perturbations and encouraging the diffusion function $\mathbf{g}(\mathbf{x}_t)$ to account for intrinsic uncertainty in the data.

**Incorporating a Denoiser Network**   To ensure that the inferred trajectories remain close to high-probability regions of the data distribution, we integrate a denoiser network into the model. Specifically, we augment the flow function with an estimate of the score function trained via denoising score matching, leading to the modified SDE:

$$d\mathbf{x}_t = \Big(\mathbf{f}(\mathbf{x}_t) + \alpha \, \nabla_{\mathbf{x}_t} \log p(\mathbf{x}_t)\Big) dt \; + \; \mathbf{g}(\mathbf{x}_t) \, d\mathbf{w}_t. \tag{21}$$

Here, we learn a denoiser network $\mathbf{d}(\mathbf{x}_t)$ to approximate $\nabla_{\mathbf{x}_t} \log p(\mathbf{x}_t)$ by training on the *denoising score matching* (DSM) objective:

$$\mathcal{L}_{\text{DSM}}(\mathbf{d}) \; = \; \mathbb{E}_{\substack{\mathbf{x} \sim p(\mathbf{x}), \\ \boldsymbol{\epsilon} \sim \mathcal{N}(\mathbf{0}, \sigma^2 \mathbf{I})}} \Big[\big\| \mathbf{d}(\mathbf{x} + \boldsymbol{\epsilon}) - \tfrac{\boldsymbol{\epsilon}}{\sigma^2} \big\|^2\Big], \tag{22}$$

where $\boldsymbol{\epsilon}$ is an isotropic Gaussian perturbation of variance $\sigma^2$. The denoiser thus learns to predict the direction pointing back toward the unperturbed sample, which acts as an approximation to the true score $\nabla_{\mathbf{x}} \log p(\mathbf{x})$. During inference, adding $\alpha \, \mathbf{d}(\mathbf{x}_t)$ to the Flow term helps guide trajectories toward regions of higher data density.

**Remark on $\alpha$:** Note that $\alpha$ need not be a fixed constant; it can be a *learnable function* of the current state $\mathbf{x}$. In principle, one might choose $\alpha(\mathbf{x}_t) = \frac{1}{2} \, \mathbf{g}(\mathbf{x}_t) \, \mathbf{g}(\mathbf{x}_t)^\top$ (or a scaled variant thereof) to *partially or fully cancel* the diffusion in the Fokker–Planck equation, balancing the expansive effect of diffusion with contraction toward high-density regions. We discuss this balance further in Section I.

**Desingularization of the Flow Loss**   The logarithmic squared loss in the flow objective (Eq. 14) has a singularity when the residual approaches zero. This singularity corresponds to a degenerate Gaussian distribution and can lead to overfitting of the flow network to the observed state changes. To mitigate this issue and enhance numerical stability, we introduce a positive regularizer $\epsilon$ into the loss function:

$$\mathcal{L}^f_{\mathbf{x}_t} = \frac{1}{2} \sum_{i=1}^{d} \log \left( \left( f_i(\mathbf{x}_t) - \frac{\Delta x_i}{\Delta t} \right)^2 + \delta \right). \tag{23}$$

This desingularization constant $\delta$ can be interpreted as a smooth transition parameter between the logarithmic square loss ($\delta = 0$) and the standard square loss ($\delta = +\infty$).

**Handling Datasets Without Explicit Time**   When dealing with datasets that lack explicit time information, we introduce a uniform, manually selected time step, $\Delta t$. In our experiments, we set $\Delta t = 1$, effectively scaling the observed changes to a suitable range where the machine learning models can converge more efficiently. This choice of $\Delta t$ influences the magnitude of the learned flow ($\mathbf{f}$) and diffusion ($\mathbf{g}$) terms. Specifically, for a scaling $\Delta t \to \lambda \Delta t$, it is straightforward to derive from Eq. 14 and 15 that the optimal flow and diffusion follow the corresponding scaling $f \to \frac{f}{\lambda}$, $g \to \frac{g}{\sqrt{\lambda}}$. This scaling, however, does not affect the underlying dynamics captured by the Neural SDE. As proven in the subsequent section, the model's inference results are invariant to the manually chosen $\Delta t$. In this sense, $\Delta t$ can be regarded as a virtual time unit which is not crucial for the learned dynamics. This property allows for training models on data without precise temporal labels.

## Proof: Temporal Scale Invariance of SDE Numerical Simulation

Consider a data-driven Stochastic Differential Equation (SDE):

$$dX = f^\theta(X)dt + g^\gamma(X)dW, \tag{24}$$

where:

- $X \in \mathbb{R}^D$ is the state variable
- $f^\theta : \mathbb{R}^D \to \mathbb{R}^D$ is the parameterized drift term
- $g^\gamma : \mathbb{R}^D \to \mathbb{R}^{D \times D}$ is the parameterized diffusion term
- $dW$ is a $D$-dimensional Wiener process increment

**Theorem 1** (Numerical Simulation Temporal Scale Invariance). *For any scaling factor $\lambda > 0$, let:*

$$\tilde{f}_d^\theta = \frac{1}{\lambda} f_d^\theta, \quad \tilde{g}_d^\gamma = \frac{1}{\sqrt{\lambda}} g_d^\gamma \tag{25}$$

*Then the scaled SDE:*

$$dX = \tilde{f}^\theta(X)(\lambda dt) + \tilde{g}^\gamma(X)dW \tag{26}$$

*has Euler-Maruyama discretization statistically equivalent to the original SDE.*

*Proof.* **Original discretization** ($\Delta t_{n,k}$):

$$X_{k+1} = X_k + f^\theta(X_k)\Delta t_k + g^\gamma(X_k)\sqrt{\Delta t_k} Z_k \tag{27}$$

**Scaled discretization** ($\lambda \Delta t_{n,k}$):

$$X'_{k+1} = X'_k + \tilde{f}^\theta(X'_k)(\lambda \Delta t_k) + \tilde{g}^\gamma(X'_k)\sqrt{\lambda \Delta t_k} Z_k \tag{28}$$

$$= X'_k + \frac{1}{\lambda} f^\theta(X'_k)(\lambda \Delta t_k) + \frac{1}{\sqrt{\lambda}} g^\gamma(X'_k)\sqrt{\lambda \Delta t_k} Z_k \tag{29}$$

$$= X'_k + f^\theta(X'_k)\Delta t_k + g^\gamma(X'_k)\sqrt{\Delta t_k} Z_k \tag{30}$$

This recovers the original discretization exactly, proving statistical equivalence. ∎ □

## E  Additional Experimental Details.

Our experiments were performed using a server on Ubuntu 22.04 LTS and 8 NVIDIA A800 80GB GPUs. The source code can be accessed at `https://github.com/NeuralSDE/NeuralSDE`.

### E.1  2D Branching Trajectories

**Multi-modal Distribution**

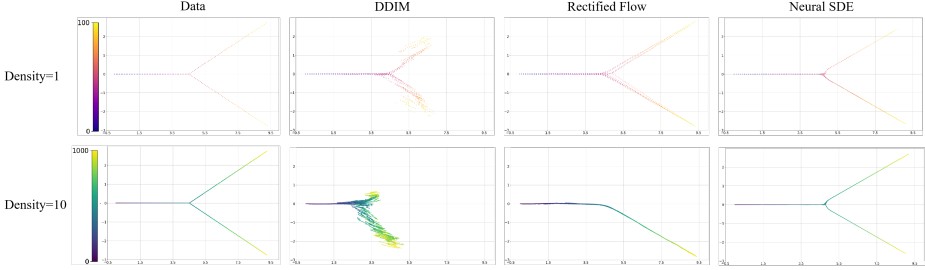

Figure 4: **Trajectory generation on a Y-shape Bifurcation (multi-modal Distribution).** We compare our proposed Neural SDE approach with DDIM and Rectified Flow at two different densities (number of steps per trajectory). At a lower density, all models successfully generate bi-modal trajectories. At a higher density, DDIM and Rectified Flow fail due to covariate-shift, while Neural SDEs still accurately captures both branches.

**Ablation Study**

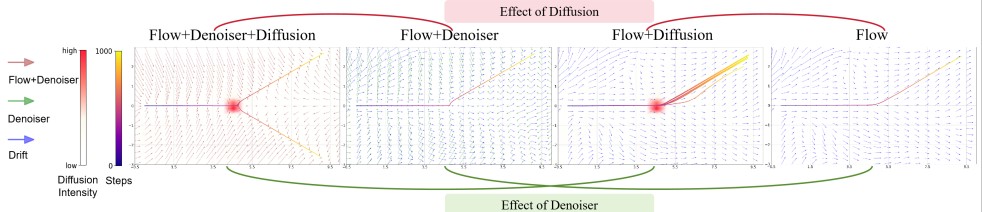

Figure 5: **Ablation Study of the Neural SDE Components on the Y-shape Bifurcation Task (high density)**. We visualize the learned vector fields with different combinations of the *Flow*, *Diffusion*, and *Denoiser* terms. The scale of vector fields is scaled for visual clarity. The *Flow* term alone captures the general direction but lacks stochasticity. Adding *Diffusion* introduces stochasticity but fails to reach the bifurcation point accurately due to covariate-shift. The *Denoiser* effectively mitigates covariate-shift. As a result, the full model (*Flow+Denoiser+Diffusion*) accurately **models** the multi-modal distribution.

**Experiment Setup**

To illustrate our method, we construct a simple synthetic dataset where the trajectory follows a piecewise constant velocity field: For time $t < 4.5$, the velocity is defined by a magnitude $u$ and an angle $\theta_1 = 0°$:

$$\frac{\mathrm{d}X}{\mathrm{d}t} = u \begin{bmatrix} \cos(\theta_1) \\ \sin(\theta_1) \end{bmatrix} = u \begin{bmatrix} \cos(0°) \\ \sin(0°) \end{bmatrix} = \begin{bmatrix} u \\ 0 \end{bmatrix}$$

After $t \geq 4.5$, the trajectories diverge, following one of two paths with angle $\theta_2 = 30°$:

$$\frac{\mathrm{d}X}{\mathrm{d}t} = u \begin{bmatrix} \cos(\theta_2) \\ \pm \sin(\theta_2) \end{bmatrix} = u \begin{bmatrix} \cos(30°) \\ \pm \sin(30°) \end{bmatrix} = \begin{bmatrix} \frac{\sqrt{3}}{2} u \\ \pm \frac{1}{2} u \end{bmatrix}$$

We generated datasets with two different densities. For the low density (visualized with $density = 1$), we sampled 100 data points. For the high density ($density = 10$), we sampled 1000 points. We trained Neural SDE, along with DDIM and Rectified Flow baselines, using the same MLP architecture to ensure a fair comparison. For each model and density configuration, we generated 10 independent trajectories.

**Network Architecture and Training Details**

For the flow, diffusion, and denoiser networks, we employ three identical MLPs. The input state $x_t$ is first transformed into a latent representation via a linear embedding layer. The main body of each network consists of five pre-activation residual blocks: each block applies LayerNorm and ReLU between two linear layers (hidden dimension 128), then adds its input to the block output before the final nonlinearity. After the residual stack, a final linear layer projects features to the output dimension; for the diffusion network, this output is then passed through a Tanh nonlinearity—separating rare non-zero responses from the mass of zeros—linearly rescaled to the target logit interval, and exponentiated to ensure positive, sharply peaked local values while preserving near-zero predictions elsewhere.

All models are trained for 200 epochs using the Adam optimizer. The initial learning rate is set to $2 \times 10^{-4}$ for the flow and denoiser networks, and $4 \times 10^{-5}$ for the diffusion network. No weight decay is applied. A `ReduceLROnPlateau` scheduler is employed for all networks, with a patience of 10 epochs and a decay factor of 0.5. The learning rate is reduced down to a minimum of $1 \times 10^{-5}$ for the flow and denoiser networks, and $1 \times 10^{-6}$ for the diffusion network. During denoiser training, Gaussian noise with a standard deviation of 0.1 is added to the input.

**E.2   Push-T**

**Non-Smooth Trajectory**

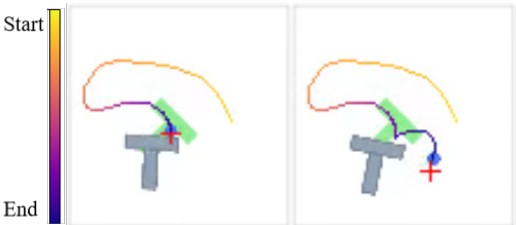

Figure 6: **Non-Smooth Trajectory Generation.** A Push-T trajectory generated by our Neural SDE, showcasing its ability to handle drastic changes in direction.

**Experiment Setup** This benchmark, initially proposed by Florence et al. and subsequently utilized by Chen et al. [Chi et al., 2023], requires controlling a circular end-effector to push a T-shaped block to a target location. The task inherently involves non-smooth trajectories due to intermittent contact and changes in contact dynamics, providing a challenging scenario for sequence modeling. For our evaluation, the agent receives as input the pose of the T-shaped block (represented by 1 keypoint and 1 angle) and the end-effector's location.

To cast imitation learning as sequence generation, we treat the expert demonstrations as sequences of states and actions. For the Diffusion Policy (DP) baseline [Chi et al., 2023], the input consists of $T_o = 2$ observations and $T_p$ noisy actions, predicting a sequence of $T_p = 16$ actions, from which $T_a = 8$ actions are used for execution. Our Neural SDE (NSDE) model concatenates the current observation and action as the state and predicts a sequence of future states. From this predicted sequence, $T_a' = 4$ actions are extracted and executed. We report results from the average of 200 environment initislization.

### Network Architecture

In this experiment, we use the same residual MLP architecture for the flow, diffusion, and denoiser modules as the 2-D case. Unless otherwise specified, we set the number of residual blocks to 4, the hidden dimension to 1024.

All networks are trained for 500 epochs using the Adam optimizer with a batch size of 512 and an initial learning rate of $2.5 \times 10^{-4}$. Distinct $L_2$ regularization values are applied to each network: $2.4 \times 10^{-6}$ for the flow network, $3.5 \times 10^{-6}$ for the denoiser, and $1.7 \times 10^{-4}$ for the diffusion network. A `ReduceLROnPlateau` scheduler is used for all models, with a patience of 20 epochs, a decay factor of 0.5, and a minimum learning rate of $1 \times 10^{-6}$. During the training of denoiser Gaussian noise with a standard deviation of 0.02 is added to the input.

## E.3 Video Prediction

| Method | FVD↓ | JEDi↓ | SSIM↑ | PSNR↑ |
|---|---|---|---|---|
| **KTH** | | | | |
| Neural SDE | $377.51 \pm 20.57$ | $1.31 \pm 0.08$ | $0.81 \pm 0.15$ | $27.89 \pm 8.62$ |
| Flow Matching | $313.82 \pm 17.16$ | $1.14 \pm 0.05$ | $0.82 \pm 0.14$ | $28.75 \pm 6.96$ |
| PFI | $312.49 \pm 12.64$ | $1.16 \pm 0.06$ | $0.82 \pm 0.14$ | $28.49 \pm 6.88$ |
| RIVER | $198.13 \pm 10.76$ | $0.54 \pm 0.05$ | $0.88 \pm 0.11$ | $31.73 \pm 6.42$ |
| **CLEVRER** | | | | |
| Neural SDE | $126.73 \pm 11.29$ | $1.06 \pm 0.06$ | $0.95 \pm 0.04$ | $34.29 \pm 8.49$ |
| Flow Matching | $99.63 \pm 3.78$ | $0.78 \pm 0.07$ | $0.95 \pm 0.04$ | $35.18 \pm 8.58$ |
| PFI | $100.31 \pm 5.51$ | $0.84 \pm 0.06$ | $0.95 \pm 0.04$ | $34.95 \pm 8.53$ |
| RIVER | $134.89 \pm 14.93$ | $1.09 \pm 0.07$ | $0.92 \pm 0.05$ | $30.49 \pm 7.71$ |

Table 2: **Performance Comparison of Video Prediction on KTH and CLEVRER.** Lower FVD and JEDi are better, while higher SSIM and PSNR are better. All metrics are computed by evaluating on 20 independent test sets, each containing 256 video sequences randomly sampled with replacement. We report the mean and standard deviation across the 20 evaluations. For PFI, Flow Matching, and RIVER, we use 100 steps, while for Neural SDE, we use adaptive step sizes with approximately 17 steps on average.

**Experiment Setup and Evaluation Protocol**

We evaluate our method on two datasets. The KTH dataset encompasses human actions performed in various real-world settings, while the CLEVRER dataset features complex interactions between simple 3D shapes governed by physical laws. For both datasets, we condition on the first 4 frames and generate 36 future frames for KTH and 12 for CLEVRER.

To ensure a fair comparison and address inconsistencies in prior work, we adopt a unified evaluation protocol as recommended by Unterthiner et al. [Unterthiner et al., 2019], due to the sensitivity of FVD to sample size. Specifically, for each dataset, we sample 256 test videos, generate one completion per video, compute the Fréchet Video Distance (FVD) against a resampled set of 256 real videos, repeat this process 20 times, and report the mean. Prior studies have reported widely varying FVD scores for the same model on the same dataset—for example, Chen et al. report 41.88 for RIVER on KTH[Chen et al., 2024], while Davtyan et al. report 180 [Davtyan et al., 2023]—despite using the same codebase and evaluation size. For consistency, we also re-evaluate RIVER using our protocol to provide a direct point of comparison.

**Network Architecture**

For the autoencoder, we use the pretrained VQGAN model provided in the repository[2], which encodes each video frame into a latent space with 4 channels. For the flow, diffusion, and denoiser networks, we adopt a U-ViT architecture based on the RIVER codebase. All three models share the same core design, which we now detail for the vector field regressor with implicit time.

We implement the network as a U-ViT [Bao et al., 2023], whose full architecture is identical across datasets. Unlike RIVER, we omit both the explicit time input $t$, the time distance $k - \tau$ and its associated context frame $x_\tau$, retaining only the state input $x_t$. Although including $t$, $k - \tau$, and $x_\tau$ can improve empirical performance, these modifications do not follow directly from a maximum-likelihood formulation, so we consider them nonfundamental and exclude them.

The state $x_t$ is first mapped into the model's latent space via a linear embedding layer. After this projection, the core of the network consists of 13 standard ViT blocks. We introduce four "long" skip connections that bridge the outputs of the first four blocks directly into the inputs of the final four blocks: at each connection point, we concatenate the feature maps channel-wise and then re-project them into the ViT hidden size. Inside each block, a LayerNorm precedes both the multi-head

---

[2]https://github.com/araachie/river

self-attention module [Vaswani et al., 2017] and the following feed-forward network. Every block employs a hidden dimension of 768 and uses eight attention heads.

All networks are trained for 300k steps with AdamW, using a learning rate of $1 \times 10^{-4}$, weight decay of $5 \times 10^{-6}$, a 5000-step linear warm-up, and a square-root decay schedule thereafter. During denoiser training, Gaussian noise (std = 0.1) is added to the input.

**Implementation Details**

We implemented Flow Matching [Lipman et al., 2022] and Probabilistic Forecasting with Interpolants (PFI) [Chen et al., 2024].[3] In the original PFI paper, the sparse conditioning mechanism from RIVER is retained—yielding performance close to RIVER. To ensure an fair comparison, we remove this sparse condition in our PFI implementation. In table 2, we provide a detailed comparison. For readers interested in the effect of sparse conditioning, we also report RIVER's results under our unified evaluation protocol; comparing those to Flow Matching highlights the effect of sparse conditioning.

In the following, we describe how each variant modifies the original RIVER inputs:

- **RIVER:** takes the reference frame $x_s$ (an interpolation of the frame $x_k$ immediately preceding the predicted frame $x_{k+1}$ and Gaussian noise), time $s$, a context frame $x_\tau$ (sampled from the past), and the interval $k - \tau$, with $\quad s \in [0, 1]$, as input.
- **Flow Matching:** removes the context inputs $\tau$ and $k - \tau$, yielding inputs $(x_s, s)$.
- **NSDE:** removes $k - \tau$, $x_\tau$, and $s$, replacing the discrete reference with a continuous state interpolation
$$x_t = (k + 1 - t)\, x_k + (t - k)\, x_{k+1}, \quad t \in [k, k+1].$$
- **PFI:** also removes $k - \tau$ and $x_\tau$, but employs a stochastic interpolant
$$I_s^k = \alpha_s\, x_k + \beta_s\, x_{k+1} + \sqrt{s}\, \sigma_s\, z_k, \quad \alpha_s = 1 - s,\ \beta_s = s^2,\ \sigma_s = 1 - s,\ z_k \sim \mathcal{N}(0, I_d), \quad s \in [0, 1].$$

All other architectural choices (U-ViT blocks, training hyperparameters, etc.) remain as described above.

In order to provide sufficient motion information for each object, all methods concatenate 4 frames to construct the state $X_t$ and generate the next frame autoregressively. If only a single frame is used, we find that the motion direction of objects cannot be determined, and as a result, the objects tend to remain stationary.
$$X_t = \left[ x_t,\, x_{t-\Delta t},\, x_{t-2\Delta t},\, x_{t-3\Delta t} \right].$$
At each training step, the model sees a pair of consecutive states $\left( X_t,\, X_{t+\Delta t} \right)$. Under the Neural SDE framework, we aim to estimate
$$\frac{dX_t}{dt} \quad \approx \quad \frac{X_{t+\Delta t} - X_t}{\Delta t}.$$
However, since $X_t$ itself encapsulates multiple consecutive frames, a naive method would be to predict *all* four finite differences:
$$\frac{1}{\Delta t} \left( x_{t+\Delta t} - x_t,\ x_t - x_{t-\Delta t},\ x_{t-\Delta t} - x_{t-2\Delta t},\ x_{t-2\Delta t} - x_{t-3\Delta t} \right).$$
Unfortunately, this can encourage a "shortcut" solution: the last three differences are merely trivial subtractions of already-visible inputs, allowing the network to ignore the first (crucial) term and simply copy known differences.

To avoid this pitfall and foster better generalization, we **only** predict the *first* difference,
$$\dot{x}_t^{(1)} = \frac{x_{t+\Delta t} - x_t}{\Delta t},$$
and rely on simple algebraic subtraction for the remaining terms:
$$\dot{x}_t^{(2)} = x_t - x_{t-\Delta t},$$
$$\dot{x}_t^{(3)} = x_{t-\Delta t} - x_{t-2\Delta t},$$
$$\dot{x}_t^{(4)} = x_{t-2\Delta t} - x_{t-3\Delta t}.$$

---

[3]All methods were adapted from the code of RIVER [Davtyan et al., 2023]. The official video-prediction code for PFI has not been released, so we re-implemented it from the paper's descriptions.

In other words, our flow network $f(\cdot)$ directly learns only the $\dot{x}_t^{(1)}$ component. By doing so, the model cannot simply "memorize" those last three differences; it must genuinely learn to predict how the next frame $(x_{t+\Delta t})$ evolves from the current frame $(x_t)$. This design choice improves generalization, as it enforces *non-trivial* predictions of future data rather than letting the network exploit short-term historical redundancy.

**Inference-Time Construction (Euler–Maruyama Step).**

At inference time, we approximate each SDE step from $t$ to $t + \Delta t$ using a single Euler–Maruyama update:

$$x_{t+\Delta t} \;=\; x_t \;+\; \Delta t\, \hat{f}(X_t) \;+\; \sqrt{\Delta t}\, \hat{g}(X_t)\, \zeta,$$

where $\zeta \sim \mathcal{N}(0, \mathbf{I})$. Once $x_{t+\Delta t}$ is computed, the state $X_{t+\Delta t}$ is updated by shifting:

$$X_{t+\Delta t} \;=\; \big[x_{t+\Delta t},\, x_t,\, x_{t-\Delta t},\, x_{t-2\Delta t}\big].$$

This procedure *respects the continuous-time Markov property*, ensuring that newly generated frames emerge directly from the learned SDE dynamics rather than from extraneous predictions of already-known differences. In doing so, the model achieves more *robust* performance for trajectory or video frame generation.

# F   High Temporal Resolution Video

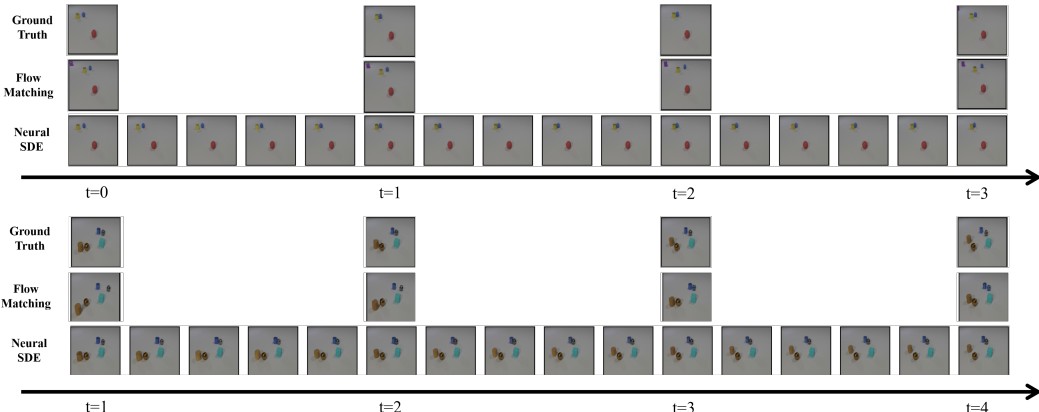

Figure 7: **High Temporal Resolution Video Generation.** This figure compares ground truth video frames with predictions from Flow Matching and our Neural SDE (NSDE) model. The ground truth frames are subsampled by a factor of 5, in order to reduce the computational cost of training. The top half of the figure shows a sequence where, in the ground truth, a yellow cylinder and a blue sphere move towards each other, appear to make contact, then move apart. Flow Matching's prediction shows the blue sphere moving towards the yellow cylinder but moving away before contact appears to occur, indicating a potential skipped frame. NSDE accurately captures the approaching, contacting, and separating motion of the two objects. The bottom half of the figure shows a sequence where, in the ground truth, a blue cylinder and a silver sphere move toward each other. Both the Ground Truth and Flow Matching show the objects moving towards each other and then separating, but missing the contact frame because of the subsampling. NSDE is able to generate intermediate frames that display the contact, demonstrating its capacity for generating videos at high temporal resolution even with relatively sparse training data. For more examples, please check out our project website https://neuralsde.github.io/.

Obtaining and training on data at high temporal resolution can be expensive. In this section, we investigate the ability of our Neural SDE model to generate videos at a high temporal resolution, even when trained on sparse, subsampled data. Figure 7 presents a visual comparison of our model's performance against a baseline, demonstrating our model's capacity to generate intermediate frames.

## G Ablation Study

| Method | FVD↓ | JEDi↓ | SSIM↑ | PSNR↑ |
|---|---|---|---|---|
| **KTH** | | | | |
| Neural SDE | **395.75** ± 23.03 | 1.325 ± 0.102 | 0.807 ± 0.148 | 27.55 ± 8.44 |
| w/o Denoiser | 408.91 ± 23.90 | **1.155** ± 0.080 | **0.828** ± 0.137 | **28.80** ± 7.37 |
| w/o Noise Injection | 470.46 ± 20.65 | 1.722 ± 0.090 | 0.787 ± 0.158 | 26.30 ± 8.98 |
| **CLEVRER** | | | | |
| Neural SDE | **128.07** ± 14.30 | **1.172** ± 0.101 | **0.944** ± 0.043 | **33.78** ± 9.19 |
| w/o Denoiser | 139.98 ± 12.76 | 1.437 ± 0.068 | 0.936 ± 0.048 | 32.63 ± 9.19 |
| w/o Noise Injection | 153.72 ± 20.11 | 1.334 ± 0.105 | 0.936 ± 0.046 | 32.82 ± 9.01 |

Table 3: **Ablation Study on Denoiser Network and Noise Injection**. Lower FVD and JEDi are better, while higher SSIM and PSNR are better. All metrics are computed by evaluating on 20 independent test sets, each containing 256 video sequences randomly sampled with replacement. We report the mean and standard deviation across the 20 evaluations.

In Table 3, we present the performance of the Neural SDE model when removing the denoiser network and the noise injection, evaluated on two video prediction datasets. The results demonstrate that both components are crucial for achieving better performance.

## H Validation Loss for Scaling Experiment

We evaluated the validation performance on the CLEVRER dataset using the reduced loss function [4] defined in Equation (31), with the desingularization constant $\delta = 0.001$ factored out:

$$\mathcal{L}' = \frac{1}{Nd} \sum_{j=1}^{N} \sum_{i=1}^{d} \log \left( \left( f_i(\mathbf{x}_t^j) - \frac{\Delta x_i^j}{\Delta t} \right)^2 + \delta \right) - \log(\delta). \tag{31}$$

## I Guiding Diffusion with a Score Function: Balancing Expansion and Contraction

In many bridging-based generative frameworks, one introduces a *score function* term to the drift of an SDE, effectively "counteracting" some portion of the diffusion. Concretely, consider modifying our baseline SDE

$$d\mathbf{X}_t = \mathbf{f}(\mathbf{X}_t) \, dt + \mathbf{G}(\mathbf{X}_t) \, d\mathbf{W}_t$$

to

$$d\mathbf{X}_t = \left[ \mathbf{f}(\mathbf{X}_t) + \alpha(\mathbf{X}_t) \nabla \ln p(t, \mathbf{X}_t) \right] dt + \mathbf{G}(\mathbf{X}_t) \, d\mathbf{W}_t, \tag{32}$$

where $\nabla \ln p(t, \mathbf{X}_t)$ is the *score function*, and $\alpha(\mathbf{X}_t)$ can be chosen as a state-dependent matrix (e.g. a diagonal function of $\mathbf{G}$). In particular, one may set

$$\alpha(\mathbf{X}_t) = \tfrac{1}{2} \mathbf{G}(\mathbf{X}_t) \mathbf{G}(\mathbf{X}_t)^\top$$

to *fully* cancel the diffusion operator in the corresponding Fokker–Planck equation (under idealized conditions). However, this extreme choice is generally not mandatory; partial or approximate cancellation can also be beneficial.

**Balancing Expansion and Contraction.** From the perspective of the probability density $p(t, \mathbf{x})$, each step in (32) comprises:

- **Diffusion**, via $\mathbf{G}$, which tends to spread or "expand" the distribution.

---

[4]See [Kaplan et al., 2020] for detailed explanation.

- **Score-based drift**, via $\alpha(\mathbf{x}) \nabla \ln p$, which pulls trajectories toward higher-density regions, acting as an "anti-diffusion" force.

By tuning $\alpha(\mathbf{x})$ appropriately, we can strike a balance between these opposing effects, preventing the distribution from diverging (if diffusion is too large) or collapsing into a narrow region (if anti-diffusion is too strong).

**Implications for Model Design.** This idea underlies many score-based generative models:

- *Adaptive Anti-Diffusion*: Rather than using a constant $\alpha$, we can learn a network that predicts $\alpha(\mathbf{x})$ based on local properties of $\mathbf{G}(\mathbf{x})$ and the data distribution.
- *Control Over Sampling Dynamics*: By partially canceling diffusion, the model can sample more efficiently—avoiding many small, iterative denoising steps—yet still capture multi-modal or uncertain behaviors where non-zero stochasticity is essential.

Hence, the controlled interplay between diffusion and a learned score function can yield flexible, stable, and computationally efficient continuous-time modeling—particularly useful for complex or high-dimensional tasks in embodied AI and generative pipelines.

