# OpenReview forum: "Neural SDEs as a Unified Approach to Continuous-Domain Sequence Modeling"
_NeurIPS.cc/2025/Conference — Submitted to NeurIPS 2025_

### Official Review · Reviewer_oK7p · 2025-06-28

**Clarity:** 1
**Significance:** 1
**Originality:** 3
**Rating:** 3
**Confidence:** 3

**Summary:**

The presented work introduces a novel algorithm to train Neural SDEs. Inspired by diffusion models and flow matching methods, the authors propose an algorithm to train SDE's without simulation.

The proposed algorithm frames the training objective as a negative log-likelihood, where the model is represented with a normal distribution, derived directly from the SDE formulation.

The authors evaluate their method in: (a) trajectory prediction problems, (b) imitation learning, and (c) video prediction.

**Questions:**

- Why do previous works learning neural SDEs require the simulation, if the aim is to match a trajectory data?
- I would encourage authors to reduce the background section to enhance readability.
- Authors should consider adding experiments to compare the performance of their method with respect to previous methods learning Neural SDEs.

**Ethical Concerns:**

["NO or VERY MINOR ethics concerns only"]

**Final Justification:**

I have increase my score from reject to borderline reject. While the authors seem confident on their work and I got a better understanding of how they are aligning the topic of deep generative models via diffusion and trajectory generation via differential models; I find it still confusing.

The writting of the paper is neither very clear and highly verbose.

**Limitations:**

yes

**Quality:**

1

**Strengths And Weaknesses:**

**Quality**
- Strenghts
The authors make a good job in finding connections between SDE's and normal distribution and exploit this relation to get a simple, yet useful training loss.

- Weaknesses
A main concern with the work is the connections the work draw between diffusion models or flow matching and their proposed Neural SDE. Despite both works make use of differential equations to represent the behavior of their models, the application and the goal are very different.

Diffusion Models, Flow Matching models or Neural ODEs aim to learn a generative model that maps a normal distribution to a target distribution, represented with the data. To aim this goal, these models represent the generative process as a differential equation. A particular relevant element of diffusion or flow matching models, in contrast with Neural ODEs is that they do not require the simulation of the diffusion process (during training) and instead are able to solve the problem for each "denoising" step.

The presented work here, to my understanding, is aiming to a complete different goal. Rather than learning a generative model using a differential equation as representation of the model, this work aims to represent stochastic dynamic processes with neural networks. Authors claim that one benefit of their proposed approach is that they avoid the simulation (similarly to flow matching avoids simulation in comparison with Neural ODEs), but it is not clear to me why simulation is needed if the intention is to learn from one-step prediction data, such as trajectories. My doubts would be: (1) Why do previous works learning stochastic dynamic processes need to apply a simulation?


**Clarity**
-Weaknesses
1. The paper provides a too long introduction before explaining what is the main contribution of the work. The authors explain their method in page 5. The paper would benefit from a restructuring such as the reader can get easier the contribution of the work.
2. I believe the connections to flow matching or diffusion models are flaw because they are essentially different problem. Despite both make use of differential equations, diffusion models aim to be generative models, while this work aims to learn a differential equation.

**Significance**
- Strenghts
The authors make a good job in representing the usefulness of their method in different tasks that could be represented as SDEs.

- Weaknesses
1. The paper would benefit from more targeted experiments. Stochastic dynamic processes are common in physics and authors could consider using data from those domains to have a more oriented experiments.
2. The paper misses comparison with previous methods learning neural SDEs.

**Originality**
- Strenghts
1. I liked that the authors exploit the relations between the normal distributions and SDEs to solve their problem.

---

> ### Author Rebuttal · Authors · 2025-07-30
>
> Thanks for appreciating our method and for raising insightful questions and concerns.
>
> 1. If the aim is to learn a trajectory, why do previous Neural SDE/ODE works require simulation?
>
> This is an intuitive question. Let's first consider the ODE case. Indeed, when learning a differential equation, the most straightforward approach is to directly learn its first-order derivative. Historically, this has been the practice in system identification. However, this approach faces significant difficulties:
>
> Accurately estimating derivatives from noisy data is inherently challenging, as differentiation amplifies noise, necessitating various filtering methods to mitigate this issue.
>
> Compounding errors: Errors from derivative fitting at single points accumulate and amplify during integration, making derivative estimation a well-known ill-posed problem (e.g. https://rss.onlinelibrary.wiley.com/doi/full/10.1111/j.1467-9868.2007.00610.x)
>
> Simulation-based methods, although also learning the derivative, mitigate these problems by penalizing the overall trajectory error, enforcing global consistency in the vector field (thus reducing compounding errors). The integration operation acts as a low-pass filter, suppressing high-frequency noise and preserving the long-term trend of the signal (e.g. https://www.sciencedirect.com/science/article/abs/pii/000510989090158E). Consequently, in practice, trajectory learning from data typically employs simulation, particularly with the advent of automatic differentiation techniques.
>
> 2. Connection with diffusion and flow is flawed since they are generative models and we are learning a differential equation.
>
> This viewpoint is not entirely accurate. While there are distinctions between generative models and trajectory generation models, they indeed share fundamental relationships. A direct piece of evidence is provided in the Neural ODE paper (https://arxiv.org/pdf/1806.07366): Section 4 discusses applying Neural ODEs for continuous normalizing flow (a generative model), whereas Section 5 discusses their use for sequence generation. The primary difference lies in the supervision signal and data coupling. In generative models, only signals at the sequence endpoint are provided, with intermediate signals unknown; thus, researchers focus primarily on the generated outcomes, forcing Neural ODE/SDE models to form probabilistic paths (many possible sequences) for generation, with data-dependent coupling (e.g., https://arxiv.org/pdf/2410.22918, page 3, last paragraph). Conversely, for time-series models, intermediate signals are available and supervision is applied throughout the entire sequence (https://arxiv.org/pdf/1806.07366, Appendix E).
>
> With the introduction of flow matching, continuous normalizing flows transitioned from simulation to simulation-free approaches (https://arxiv.org/pdf/2210.02747, section 5). However, in time-series modeling, simulation remains necessary. In our work, we view trajectory segments within specific intervals as transport processes, borrowing training strategies of stochastic interpolants with data-dependent coupling (https://arxiv.org/pdf/2310.03725, Algorithm 1). This results in learned velocities composed of two components: the first-order derivative derived from interpolation and the score function. Our derived maximum likelihood objective from the SDE transition density contains only the first-order derivative. Considering the score function captures global distribution information, it compensates for the compounding error arising from directly learning the first-order derivative alone, motivating our addition of a denoiser.
>
> 3. Introduction and background are too long.
>
> We acknowledge this concern. However, fully appreciating the context and significance of our work necessitates broad background knowledge, including familiarity with flow matching, diffusion models, Neural SDE/ODE generative models, time-series modeling, control theory, system identification, and their historical developments and interconnections. Historically, direct derivative learning faced challenges from noise-induced estimation difficulties and compounding errors, prompting the development of filtering methods and later integral-based simulations. Inspired by denoising score matching, our approach applies a denoiser to reduce compounding errors, effectively mitigating the limitations of derivative-based methods and validating direct derivative learning once again. We will revise our introduction and background sections to better convey this historical context to readers.
>
> 4. Add experiments on physical datasets and compare with previous Neural SDE methods.
>
> This is an excellent suggestion. We will incorporate comparative experiments with simulation-based Neural SDE methods (as described in https://arxiv.org/pdf/2001.01328, section 7.3).

---

> > ### Comment · Area_Chair_RdC4 · 2025-08-04
> >
> > Dear Reviewer,
> >
> > Please respond to the rebuttal. Thanks.
> >
> > AC.

---

> > ### Comment · Reviewer_oK7p · 2025-08-04
> > **Response to the Author**
> >
> > Thank so much for the provided response.
> >
> > The provided answer helped me understand the view of the authors. I agree on the view that the difference between Neural ODE/SDE applied for generative modelling and applied for trajectory prediction is based on the type of data there is available:
> >
> > In generative models, you have access to the final point and the intermediate points are "estimated" rather implicitly (simulation-based) or given explicitly (simulation-free) in the style of diffusion/flow matching models. In particular, in simulation-free approaches, the intermediate signal is used in the loss function.
> >
> > In contrast, in trajectory generation, you assume access to the intermediate points. They are explicitly use in the loss function. Additionally, previous methods apply simulation to reduce compounding errors.
> >
> > - Can you help me understand. Is your method, in contrast with the baselines, not using simulation to induce global consistency/avoiding compounding errors?
> >
> >
> > Thanks again for the clarity of your response.

---

> ### Author Response · Authors · 2025-08-05
>
> Thank you for your reply. Our method indeed does not use simulation. We use a synergistic strategy composed of three core components that work together to ensure global consistency and mitigate compounding errors, as discussed in detail in Appendix D and evidenced by our experimental results in Figure 5 and Appendix G.
>
> These three key components are:
>
> Denoiser Network (Global Distribution Correction): It is well-established in the literature that for modern generative models, the drift of the learned dynamics is composed of two key parts. For frameworks like Flow Matching and Stochastic Interpolants, the drift of the corresponding ODE implicitly contains a score function component (https://arxiv.org/pdf/2011.13456 section4.3 https://arxiv.org/pdf/2303.08797 figure 3). The drift of the corresponding SDE in diffusion and Stochastic Interpolants (https://arxiv.org/pdf/2011.13456 section4.2 https://arxiv.org/pdf/2303.08797 figure3) contains this score function explicitly. In the context of generative modeling, where supervision is typically limited to the endpoints of the sequence (e.g., noise-to-data), there are various design choices for the other, non-score part of the drift (https://arxiv.org/pdf/2011.13456 Appendix B). However, our key insight is that for sequential modeling, where intermediate trajectory states are available, we can leverage this rich supervisory signal to learn this non-score component directly from data. This is precisely what our framework achieves: the flow term $f(x)$ is derived directly from a Maximum Likelihood objective on the observed transitions ($x_t, x_{t+Δt}$). This data-driven term learns the deterministic part of the dynamics, while we explicitly add the denoiser network. As we discussed, the denoiser learns global information about the entire data distribution. During inference, it provides a powerful global corrective force that "pulls back" any trajectory that deviates from the data manifold due to the accumulation of local errors. This is our direct mechanism for ensuring global consistency .
>
> State Interpolation (Dense Local Supervision): Traditionally, estimating derivatives directly from sparse observations $(x_{t_{k}},\ x_{t_{k+1}})
> $ leaves the intermediate dynamics undefined, which can lead to locally inaccurate velocity fields that generate and compound errors. By interpolating between observed states during training, we generate a multitude of intermediate points $x_\tau \quad (\tau \in [t_k,\ t_{k+1}])$. This provides the model with a much denser supervisory signal along the trajectory path, forcing the learned flow field $f(x)$ to be smoother and more self-consistent locally.
>
> Noise Injection (Guaranteed Model Smoothness): This is a crucial aspect of our method that provides a theoretical guarantee for mitigating compounding errors. We inject Gaussian noise $\eta \sim \mathcal{N}(0,\ \sigma^2 I)$ into the interpolated states. This is more than just an ad-hoc data augmentation. As rigorously demonstrated in the Stochastic Interpolants framework (https://arxiv.org/pdf/2303.08797 theorem 2.6 & Appendix B.1), the introduction of this latent noise variable (termed $\gamma(t)z$ in their work) mathematically acts as a smoothing operator. Specifically, this noise term introduces a Gaussian decay in the characteristic function of the interpolant's probability distribution. This property guarantees that the learned velocity field $b(t,x)$ and score $s(t,x)$ are spatially smooth. A smooth velocity field is essential for suppressing compounding errors because it is inherently robust to small perturbations; it mitigates the amplification of minor numerical errors during integration.
>
> This combined "smooth model + local accuracy + global correction" approach allows us to robustly learn the dynamics without simulation.

---

### Official Review · Reviewer_G7qB · 2025-06-30

**Clarity:** 2
**Significance:** 2
**Originality:** 3
**Rating:** 4
**Confidence:** 3

**Summary:**

This paper considers sequence learning via Neural SDEs using drift and diffusion terms. Instead of the standard training scheme that involves simulating the SDE, it proposes a maximum-likelihood training scheme using a simulation-free approach. Assuming dense, noise-free observed trajectories, it avoids explicit SDE simulation by directly modeling the one-step transition distribution $p(x_{t+1} \vert x_t)$ based on Euler discretization given $x_t$. The Gaussian likelihood is further decomposed into two terms that separate the effects of drift learning and diffusion learning. Further assuming the SDE is time-invariant and the drift term is diagonal for efficient likelihood computation, a decoupled training strategy is introduced to improve performance: the drift function is first fitted by minimizing a log-squared residual loss, followed by estimating the diffusion term via residual matching. The approach also enjoys fast sampling by parallel sampling segement to approximate a SDE trajectory

demonstrates effectiveness on imitation learning and video prediction tasks, emphasizing low inference cost compared to flow-matching and diffusion-based models.

**Questions:**

- There are also works that leverage dense observed trajectories and Euler–Maruyama transition likelihoods for training SDEs (e.g., [1]). Since this is not discussed in the paper, how is the proposed approach similar to and different from such prior work?
- Since incorporating time only affects training complexity but does not fundamentally alter the pipeline, the motivation for choosing a time-invariant model is not entirely clear to me. Could the authors provide some comments on this aspect?

[1] https://arxiv.org/abs/2105.08449

**Ethical Concerns:**

["NO or VERY MINOR ethics concerns only"]

**Final Justification:**

During the rebuttal phase, the authors clarified my main confusion regarding the sequence modeling aspect, addressing a key concern and leading me to increase my score. However, I still recommend a careful revision to improve clarity, and I remain slightly concerned that such modifications may introduce noticeable changes to the updated paper. Therefore, I maintain a borderline accept recommendation.

**Limitations:**

The aforementioned weakness should listed as limitations until further clarification is provided.

**Paper Formatting Concerns:**

A minor yet noticable issue is it is not using under review format but accept format, this makes the line label missing.

**Quality:**

2

**Strengths And Weaknesses:**

Strengths:
- The two-stage optimization (first drift, then diffusion) seems interesting and improves training stability, especially compared to jointly optimizing drift and diffusion terms.

Weaknesses:
- Assumptions: From my perspective, this approach relies heavily on strong assumptions (access to dense, noise-free trajectories where each state is fully observed), which is rarely the case in real-world scenarios. This limits its general applicability.
- Writing: This is the main drawback of the paper. I think many parts are not clearly written and should be rephrased for clarity and correctness. For instance:
   - The claim of the second contribution is rather confusing or even misleading. First, it is unclear how this method accelerates sampling compared to training—using Euler–Maruyama discretization on a standard SDE results in the same number of function evaluations (NFE) per step. Second, diffusion or flow-matching models require simulation because (1) they must generate samples from new initial conditions, and (2) they do not observe intermediate states, unlike sequence modeling tasks. This paper considers a completely different setting, where the trajectory is already observed, and thus acceleration is only possible under this assumption via conditional modeling. This is not an inherent advantage over generative models—such a claim, in my view, is incorrect.
    - In Sec. 2.1.1, diffusion models are known to be discrete in their original Markov chain form, but the term “discrete diffusion model” is more commonly used to refer to models with discrete state spaces rather than discrete time.
    - After Eq. 1: I am not sure if “flow coefficient” is a standard term in the SDE literature.
    - Eq. 4 is not inherently tied to a discrete formulation.

Given the above, unless further clarification is provided, I would suggest a careful rewrite of these sections for accuracy and clarity.

---

> ### Author Rebuttal · Authors · 2025-07-31
>
> Thanks for your review and thoughtful comments.
>
> On the Assumption of Dense, Noise-Free Trajectories
>
> We appreciate this insightful concern; however, it stems from a misunderstanding. The concern likely originates from the known difficulties in accurately estimating trajectory derivatives from noisy data, as differentiation amplifies noise and integral computations compound errors—a well-known ill-posed problem. This challenge is precisely why recent sequence modeling methods rely on simulation-based techniques. Integration inherently functions as a low-pass filter, effectively suppressing high-frequency noise while preserving the long-term trends of signals.
>
> Our work is not ignorant of this challenge; it is explicitly designed to solve it without resorting to simulation. Our practical implementation robustly handles realistic, sparse, and noisy data through a three-part strategy:
>
> Handling Sparsity: We use a state interpolation strategy during training , a common technique in flow-matching frameworks. This effectively "densifies" the training data, providing a continuous signal for the model to learn from.
>
> Robustness to Noise: We employ noise injection during training, as detailed in Appendix D . Perturbing the states with Gaussian noise regularizes the model, making it robust to the noisy observations found in real-world data.
>
> Mitigating Compounding Error: We introduce a denoiser network inspired by denoising score matching . The denoiser approximates the score function, which captures global distribution information. This global guidance regularizes the learned dynamics and mitigates the accumulation of local errors—the very problem that makes direct derivative learning ill-posed.
>
> On the Clarity of Writing and Contributions (Inference Efficiency)
>
> We appreciate your recognition that our setting differs from the prevalent sequence prediction paradigm.
>  Many modern sequence prediction models (e.g., diffusion policy) employ a nested, two-timescale process:
> An outer, auto-regressive loop proceeds in discrete "real-time" steps (e.g., frame-by-frame).
> For each step, an inner loop runs a full generative process in "pseudo-time," transporting a sample from a simple prior (noise) to the data manifold (the next frame). This is a "long" transport path with a high NFE count.
>
> Our approach unifies these two timescales. We directly model the dynamics along the single, continuous, real-time axis. This is a more direct and natural representation for continuous-time phenomena.
>
> Indeed, Euler-Maruyama integration incurs the same Number of Function Evaluations (NFE) per step across models. Our advantage stems not from per-step costs but from significantly fewer steps required to generate trajectories.
>
> The crucial distinction is total transport cost:
>
> Diffusion/Flow-Matching Models for sequence generation transport samples from simple priors (pure noise) to data manifolds, inherently long paths necessitating high NFEs.
>
> Our Neural SDE model transports locally from the previous observed state  to the next state . The proximity of start and end points substantially reduces required integration steps.
>
> We also acknowledge and will correct terminology for clarity: "discrete diffusion model" will be specified as "discrete-time," and "flow coefficient" as the more standard "drift term/coefficient," as recommended.
>
> Comparison to Related SDE Training Methods
> Thank you for the reference. Our method differs from such works in several key aspects:
>
> Modeling of Time and Dynamics: The cited work relies on fixed time steps defined by the dataset's sampling frequency. Our method learns a continuous-time SDE, which allows for flexible simulation at any temporal resolution during inference. This enables capabilities like "free" temporal interpolation, which we demonstrate in Appendix F .
>
> Mitigation of Compounding Error: A core novelty of our work is the explicit mitigation of compounding error via a denoiser that injects global distributional information. To our knowledge, prior works that use local Euler-Maruyama likelihoods do not incorporate such a mechanism to regularize the learned dynamics.
>
> Optimization Strategy: We introduce a decoupled two-stage optimization of drift and diffusion . This enhances training stability and avoids the significant challenge of computing and inverting a high-dimensional covariance matrix, which can be a bottleneck in methods that jointly optimize the likelihood.
>
> We will add a discussion of this class of methods to our related work section to better situate our contributions.
>
> Motivation for a Time-Invariant Model
>
> Our choice of a time-invariant model is a principled one based on the concept of an autonomous system in mathematics(https://en.wikipedia.org/wiki/Autonomous_system_(mathematics)), where the governing dynamics depend only on the system's state, not explicitly on time. It is important to note that the model is still implicitly time-dependent. The drift and diffusion are functions of the state x, which itself evolves continuously in time. Therefore, the system's behavior naturally changes over time. While explicitly adding time as an input, is a straightforward extension for engineering better performance on specific tasks, our goal was to first develop a principled and general framework.

---

> > ### Comment · Area_Chair_RdC4 · 2025-08-04
> >
> > Dear Reviewer,
> >
> > Please respond to the rebuttal. Thanks.
> >
> > AC.

---

> > > ### Comment · Area_Chair_RdC4 · 2025-08-05
> > >
> > > Dear Reviewer,
> > >
> > > Please comment on the rebuttal. Thanks.
> > >
> > > AC.

---

> > > > ### Comment · Reviewer_G7qB · 2025-08-07
> > > >
> > > > I thank the authors for their detailed rebuttal, which has clarified my earlier confusion regarding the claims of the paper. I believe a major source of this confusion comes from the presentation: the paper could more clearly explain its use of generative SDEs for sequence modeling, rather than making it sound like an acceleration of standard diffusion/flow-matching approaches. I note this appears to be a general issue also reflected in other reviewer comments. I find the work itself very interesting, but it could benefit greatly from a clearer presentation of the sequence modeling aspect to avoid such confusion and to better highlight its contribution, which would substantially strengthen the paper.

---

> > > > > ### Author Response · Authors · 2025-08-08
> > > > >
> > > > > Thank you for your recognition of our work. We have also noticed the same issue and appreciate your feedback. We will carefully summarize the points of confusion raised during this discussion and improve our presentation accordingly in the camera-ready version to more clearly convey the sequence modeling aspect of our method. We sincerely hope you could also share your positive assessment of our work with the other reviewers.

---

> ### Author Response · Authors · 2025-08-05
>
> To be more specific, our method's robustness to sparse and noisy data is not an ad-hoc fix but a core design principle, achieved through the synergy of three components detailed in Appendix D and evidenced by our experimental results in Figure 5 and Appendix G.
>
> Dense Local Supervision via State Interpolation: By interpolating between observed states during training , we provide a dense supervisory signal. This forces the learned flow field to be locally accurate and self-consistent, preventing inaccuracies from arising between sparse observations.
>
> Guaranteed Model Smoothness via Noise Injection: As theoretically grounded in the Stochastic Interpolants framework (https://arxiv.org/pdf/2303.08797 theorem 2.6 & Appendix B.1), it acts as a smoothing operator on the learned dynamics. This guarantees a spatially smooth velocity field, which is crucial for suppressing the amplification of small integration errors.
>
> Global Consistency via a Denoiser Network: It is well-established in the literature that for modern generative models, the drift of the learned dynamics is composed of two key parts. For frameworks like Flow Matching and Stochastic Interpolants, the drift of the corresponding ODE implicitly contains a score function component (https://arxiv.org/pdf/2011.13456 section4.3 https://arxiv.org/pdf/2303.08797 figure 3). The drift of the corresponding SDE in diffusion and Stochastic Interpolants (https://arxiv.org/pdf/2011.13456 section4.2 https://arxiv.org/pdf/2303.08797 figure3) contains this score function explicitly. In the context of generative modeling, where supervision is typically limited to the endpoints of the sequence (e.g., noise-to-data), there are various design choices for the other, non-score part of the drift (https://arxiv.org/pdf/2011.13456 Appendix B). However, our key insight is that for sequential modeling, where intermediate trajectory states are available, we can leverage this rich supervisory signal to learn this non-score component directly from data. This is precisely what our framework achieves: the flow term $f(x)$ is derived directly from a Maximum Likelihood objective on the observed transitions ($x_t, x_{t+Δt}$). This data-driven term learns the deterministic part of the dynamics, while we explicitly add the denoiser network. The denoiser learns the global data distribution and provides a powerful corrective force during inference by approximating the data's score function. It pulls any trajectory that begins to deviate back towards the data manifold, directly ensuring global consistency and mitigating the accumulation of local errors over long sequences.
>
> This combination of ensuring local accuracy, guaranteeing model smoothness, and enforcing global correction is precisely what allows our framework to robustly handle realistic data without resorting to simulation-based training.

---

### Official Review · Reviewer_yZgJ · 2025-07-01

**Clarity:** 3
**Significance:** 1
**Originality:** 1
**Rating:** 4
**Confidence:** 4

**Summary:**

This paper proposes a neural stochastic differential equation (Neural SDE) framework for modeling continuous-domain sequential data. The authors derive a maximum likelihood training objective based on the Euler–Maruyama discretization and propose a decoupled two-stage optimization strategy for learning the drift and diffusion components of the SDE. The model is evaluated on different tasks.

**Questions:**

1. On Equation (13): Could the authors clarify how they justify this equation? Since $\Delta x$ is a random variable, its square divided by $\Delta t$ cannot equal the deterministic function $\sigma^2(x_t)$ unless additional assumptions are made or an expectation is taken.

2. On novelty: How does your approach differ substantially from existing work that uses Euler–Maruyama-based likelihood estimation for SDEs? For example, how does it compare to classical approaches from the statistics literature or prior Neural SDE works?

3. On missing theoretical motivation: Equations (14) and (15) appear to rely on a moment-based interpretation of the SDE transition kernel. Why did the paper not derive them directly from first and second moment identities of the SDE transition distribution?

4. Could your approach be viewed as a special case of existing methods (e.g., score-based models or variational inference over paths)? If not, what exactly sets your framework apart?

**Ethical Concerns:**

["NO or VERY MINOR ethics concerns only"]

**Final Justification:**

Thank authors for their rebuttal and discussion. In particular, their clarifications have helped the reviewers better understand the theoretical framework of the paper. In my view, the work would benefit from a stronger focus on SDE modeling, along with more targeted experimental design and evaluation, which would also enhance the theoretical clarity of the paper.

**Limitations:**

Yes

**Paper Formatting Concerns:**

No major formatting issues were found.

**Quality:**

2

**Strengths And Weaknesses:**

Strengths:

The paper is well-written and the experiments are carefully executed.
The idea of combining a decoupled estimation of drift and diffusion terms in the context of Neural SDEs is intuitively appealing.
The authors benchmark their method on a variety of tasks, and the results are competitive with existing methods in terms of both quality and efficiency.

Weaknesses:

Lack of novelty: The core methodology—using maximum likelihood estimation based on Euler–Maruyama discretization to fit SDE parameters—is standard and well-documented in the literature. For instance, the technique is discussed in detail in works such as Joseph Jesiman’s PhD thesis ([https://eprints.qut.edu.au/16205/](https://eprints.qut.edu.au/16205/)). The paper does not clearly articulate how it substantially differs from prior works using similar ideas.

Incorrect derivation of key result: The most seemingly novel element in the paper is Equation (13), which expresses the optimal diffusion as the squared residual of the drift term. However, this equation is fundamentally flawed: the right-hand side involves a random variable (Δx), whereas the left-hand side is deterministic for a given $x_t$. Equating the two is mathematically incorrect.

Misinterpretation of moment-matching structure: Equations (14) and (15) appear to be motivated by a moment-matching intuition derived from the known first and second moments of SDE transition distributions. (See: https://journals.aps.org/pre/pdf/10.1103/PhysRevE.106.014140 ) However, the authors do not acknowledge or derive these results from that perspective, suggesting a lack of awareness of the theoretical basis underlying their method.

Clarity of novelty: The presentation does not distinguish the proposed approach sufficiently from prior work on training Neural SDEs. The contribution is framed as "simulation-free maximum likelihood" training, but this is a common and natural strategy in the SDE literature.

Overall: While the paper is well-executed in terms of engineering and experimental design, the theoretical contribution is either incorrect (Eq. 13) or lacks novelty. Thus, I cannot recommend acceptance.

---

> ### Author Rebuttal · Authors · 2025-07-31
>
> We sincerely thank the reviewer for their detailed feedback and the opportunity to clarify our contributions. The reviewer's core concerns appear to stem from a misunderstanding of our core technical motivation and the positioning of our work. We will address each point systematically.
>
> 1. On the Justification and Correctness of Equation (13)
> This is the most critical point. We respectfully clarify that Equation (13) is not a probabilistic identity but rather the analytical minimizer of the single-sample Negative Log-Likelihood (NLL) with respect to the diffusion term $g_i(x_t)$. Your interpretation that we are equating a deterministic function with a random variable is a misunderstanding of this optimization step.
>
> Our starting point is the single-transition NLL in Equation (11), which is a function of both the drift $f$ and the diffusion $g$. This equation correctly represents the log-likelihood of observing a transition from $x_{t + \Delta t}$ given the SDE parameters.
>
> To derive our decoupled optimization scheme, we first find the optimal value for the diffusion term, $g(x_t)$, that minimizes this NLL for a given data sample $(x_t, \Delta x, \Delta t)$ and a fixed drift function $f$.
>
> By taking the partial derivative of $\mathcal{L} _{x_t}$ in Eq. (11) with respect to $\sigma_i^2(x_t)$ and setting it to zero, we arrive at the expression in Equation (13). This equation defines the value of $\sigma_i^2(x_t)$ that would perfectly explain the observed residual $\left( f_i(x_t) - \frac{\Delta t_i}{\Delta x_i} \right)
> $ as stochastic noise for that specific sample.
>
> This analytical solution then motivates the objective function for the diffusion network in Equation (15). We are not claiming Eq. (13) holds for all possible $\Delta x$; rather, we are training a network $\sigma_\theta(x_t)$ to regress towards this optimal, sample-dependent value.
>
> In summary, the derivation is mathematically sound and follows a standard procedure for deriving two-stage or conditional optimization objectives. We apologize if our presentation was unclear and will revise Section 3.2 to explicitly state that Eq. (13) is an analytical minimizer, not a probabilistic identity, to prevent future misunderstanding.
>
> 2. On the Novelty of Our "Simulation-Free" Framework
> We agree that Maximum Likelihood Estimation for SDEs is a well-established concept. However, our novelty does not lie in inventing MLE, but in developing a framework that makes a simulation-free, direct-derivative learning approach practical and effective for complex, high-dimensional sequence modeling tasks like video prediction, a domain where this was previously considered infeasible.
>
> Classical SDE literature and prior Neural SDE works often avoid direct, simulation-free derivative fitting for two main reasons:
>
> Noise Amplification: Directly estimating derivatives from discrete, often noisy, data is an ill-posed problem.
>
> Compounding Error: Local errors in the learned vector field accumulate during integration (inference), leading to divergent trajectories. This is precisely why most modern methods for sequence modeling (e.g., other Neural SDEs) rely on costly forward simulations during training, as the integration step enforces global consistency and filters noise.
>
> Our core contribution is a novel framework that mitigates this fundamental challenge. The novelty rests on two key pillars:
>
> Explicit Error Mitigation via a Denoiser: Our contribution is the introduction of a denoiser network, which approximates the score function of the data distribution $\left( \nabla_{x_t} \log p(x_t) \right)
> $, into the drift term (Eq. 21). This term injects global distributional information into the local, derivative-based learning process. As shown in our ablation studies (e.g., Figure 5), this denoiser is crucial for mitigating the covariate shift and compounding errors that cause simpler models to fail, especially in multi-modal scenarios. To our knowledge, this explicit mechanism to regularize the learned dynamics and make simulation-free training viable is novel.
>
> Stable Decoupled Optimization: We introduce a decoupled two-stage optimization of drift and diffusion . This enhances training stability and avoids the significant challenge of computing and inverting a high-dimensional covariance matrix, which can be a bottleneck in methods that jointly optimize the likelihood.
>
> Therefore, our work should not be seen as just another application of a classical method, but as a new paradigm that makes a classically difficult approach (simulation-free learning) work in modern, complex domains by integrating key ideas from recent generative models.
>
> 3. On Theoretical Motivation and Connection to Moment-Matching
> We thank the reviewer for pointing out the insightful connection to moment-matching. While our objective functions for the drift (Eq. 14) and diffusion (Eq. 15) can indeed be interpreted through this lens, our derivation is grounded directly and explicitly in the principle of Maximum Likelihood Estimation from the Gaussian transition density of the SDE.
>
> We chose this ML-based derivation because it provides a single, unified, and rigorous theoretical origin for our entire framework. The simplified objectives in Eqs. (14) and (15) are not ad-hoc choices but are the direct consequences of our proposed decoupled optimization scheme applied to the NLL. We believe this derivation provides a clearer and more principled foundation for our specific two-stage algorithm.
>
> We agree this connection is valuable for interpretation and will add a brief discussion to the final version, acknowledging that our ML-derived objectives are consistent with the principles of matching the first and second moments of the transition kernel.
>
> 4. Positioning With Respect to Other Methods (e.g., Score-Based Models)
> We thank you for this question, as it allows us to clarify the precise positioning and novelty of our work at the intersection of generative modeling and sequential modeling.
>
> First, we'd like to emphasize that Neural ODEs/SDEs are a foundational framework applicable to both domains. As the original Neural ODE paper(https://arxiv.org/pdf/1806.07366) demonstrates, the same tool can be used for continuous normalizing flows (a generative task) and time-series modeling (a sequential task). The key distinction lies in the supervision: generative models often work with unpaired data or only endpoint conditions, while sequential models are supervised on intermediate states along a trajectory.
>
> A critical divergence occurred in how these two fields evolved:
>
>
> Generative Modeling: With the advent of Flow Matching and Stochastic Interpolants, the field successfully transitioned to
> simulation-free training. These methods learn a vector field to transport between distributions without costly simulations.
>
> Sequential Modeling: This domain has remained largely dependent on simulation-based training. The reason is that directly fitting a derivative from local data $\left( \frac{\Delta x}{\Delta t} \right)$ is a notoriously ill-posed problem, suffering from noise amplification and compounding errors during inference. Simulation acts as a regularizer, enforcing global consistency on the learned vector field.
>
> Our work is the first to successfully bridge this gap, introducing a practical simulation-free framework for Neural SDEs in the context of sequential modeling.
>
> Our key insight comes from analyzing modern simulation-free generative methods. Specifically, stochastic interpolants with data-dependent couplings(https://arxiv.org/pdf/2310.03725, Algorithm 1) show that an optimal transport velocity can be decomposed into two essential components:
>
> A simple first-order derivative derived from interpolating between data points.
>
> A score function term that captures global distributional information.
>
> This is precisely where our framework its novel synthesis:
>
> Our maximum likelihood derivation for the flow term (Eq. 14) naturally and coincidentally arrives at an objective for learning the first-order derivative component.
>
> However, we recognize that this alone is insufficient. The historical failures of direct derivative fitting are due to the absence of the second component.
>
> Our crucial contribution is to explicitly add a denoiser network to the drift term (Eq. 21, 440). This denoiser is trained to approximate the score function$\left( \nabla_{x_t} \log p(x_t) \right)$ and provides the missing global regularization. It is this component that stabilizes the learning process and counteracts compounding errors, finally making direct, simulation-free derivative learning viable for complex, high-dimensional sequences

---

> > ### Comment · Reviewer_yZgJ · 2025-08-06
> >
> > Thank you very much for the authors’ detailed responses. My current concerns focus on the following points:
> >
> > 1. Regarding Equation (13), I appreciate the clarification and now understand the authors' intended meaning. However, I still find the current notation unsatisfactory—especially when comparing Equations (13) and (15). I suggest introducing new symbols for g_i and sigma_i in Equation (13) to avoid potential confusion. In addition, is the logarithm in Equation (14) a typo? If the log is removed, it should be possible to prove that the estimation procedure defined by Equations (14)–(15) is consistent in the limit of infinite data and sufficient model capacity. With the log present, however, this consistency is unclear and may not hold.
> >
> > 2. Related to the above, parameter estimation for SDEs using Euler–Maruyama discretization and maximum likelihood methods is already a well-studied area. If this paper contributes to that line of work, the novelty seems primarily technical, and possibly minor. Moreover, the manuscript lacks empirical comparisons with existing SDE parameter estimation methods, which makes it difficult to assess the actual advancement over prior work.
> >
> > 3. Concerning the “simulation-free” claim, I would like to point out that many classical approaches—including neural SDEs—require simulation during training largely because Euler–Maruyama approximations break down at large lag times. In contrast, this paper handles long lag times using simple linear interpolation. I am not convinced that such a straightforward approach yields valid modeling results, and the authors provide little justification or analysis regarding the reasonableness of this choice.

---

> ### Author Response · Authors · 2025-08-05
>
> We respectfully clarify that our framework is specifically designed to overcome the classic challenge of compounding errors inherent in simulation-free training. This is achieved not through a single mechanism, but a synergistic strategy of three core components, which are detailed in Appendix D and empirically validated in our ablation studies:
>
> A Denoiser Network for Global Correction: It is well-established in the literature that for modern generative models, the drift of the learned dynamics is composed of two key parts. For frameworks like Flow Matching and Stochastic Interpolants, the drift of the corresponding ODE implicitly contains a score function component (https://arxiv.org/pdf/2011.13456 section4.3 https://arxiv.org/pdf/2303.08797 figure 3). The drift of the corresponding SDE in diffusion and Stochastic Interpolants (https://arxiv.org/pdf/2011.13456 section4.2 https://arxiv.org/pdf/2303.08797 figure3) contains this score function explicitly. In the context of generative modeling, where supervision is typically limited to the endpoints of the sequence (e.g., noise-to-data), there are various design choices for the other, non-score part of the drift (https://arxiv.org/pdf/2011.13456 Appendix B). However, our key insight is that for sequential modeling, where intermediate trajectory states are available, we can leverage this rich supervisory signal to learn this non-score component directly from data. This is precisely what our framework achieves: the flow term $f(x)$ is derived directly from a Maximum Likelihood objective on the observed transitions ($x_t, x_{t+Δt}$). This data-driven term learns the deterministic part of the dynamics, while we explicitly add the denoiser network. The denoiser is trained to approximate the score function of the data distribution, thereby learning global information. During inference, it provides a powerful corrective force that pulls any trajectory that begins to diverge back towards the high-density regions of the data manifold. This component is our primary mechanism for ensuring global consistency and is crucial for preventing trajectory divergence, as shown in our bifurcation experiment.
>
> State Interpolation for Dense Local Supervision: By densely interpolating between observed states during training, we provide the model with a much richer supervisory signal along the trajectory path. This forces the learned flow field to be smoother and more accurate between the discrete observations, significantly reducing the local inaccuracies that would otherwise initiate and amplify compounding errors.
>
> Noise Injection for Guaranteed Model Smoothness: We inject Gaussian noise into the interpolated states as rigorously shown in the stochastic interpolant (https://arxiv.org/pdf/2303.08797 theorem 2.6 & Appendix B.1), this acts as a mathematical smoothing operator. It guarantees that the learned velocity field is spatially smooth, making it inherently more robust to small perturbations and suppressing the amplification of numerical errors that occurs during integration.
>
> This combined "global correction + dense local accuracy + guaranteed model smoothness" approach provides a robust and principled foundation that makes simulation-free derivative learning viable for complex, high-dimensional sequences.

---

> ### Author Response · Authors · 2025-08-06
>
> We sincerely thank you for your continued engagement. We would like to offer the following clarifications.
>
> 1. Regarding Equations (13), (14), and their consistency:
>
> a. We accept the suggestion to improve our notation. In the final manuscript, we will add asterisks to the terms in Equation (13) (i.e.,  $g_i^\ast$ and $\sigma_i^\ast$) to explicitly denote that they are the analytical minimizers.
>
> b. We wish to re-emphasize that the logarithm in Equation (14) is not a typo. It is derived directly by substituting the minimizer for the diffusion term from Equation (13) back into the full NLL objective in Equation (11).
>
> c. In terms of consistency, we will provide a formal proof in the appendix of the revised manuscript.
>
> 2.Regarding the novelty and contribution of our work:
>
> We respectfully argue that our contribution is not minor. To the best of our knowledge, no prior work has made direct SDE parameter estimation effective for high-dimensional and complex data like video. Our core contribution is analyzing the success of modern generative frameworks like Flow Matching, identifying the key components that enable them to work as detailed in our previous comment, and successfully adapting these principles to the direct SDE estimation paradigm. This synthesis is what makes a classical approach viable in a domain where it previously was not. We believe this constitutes a significant practical and conceptual contribution. To further contextualize our work, we will add a comparison against a classical direct SDE estimation method on a low-dimensional task. Our primary goal is to show the deep learning community that with the appropriate framework, direct SDE estimation is a powerful and viable approach for their tasks. We aim to show that for sequential tasks, one can bypass the "from-noise-to-data" paradigm of generative models and instead use a more direct and efficient approach.
>
> 3. Regarding linear interpolation:
>
> a. Using linear interpolation during training does not imply the learned result will be linear. There is clear difference between the conditional velocity field and the marginal velocity field (see Figure 3 in https://arxiv.org/pdf/2412.06264). It is possible to raise question that these conclusions are from generative models, where data pairs aren't initially coupled, whereas trajectory datasets inherently contain paired trajectory segments. If you also hold this concern, please refer to https://arxiv.org/pdf/2310.03725.
>
>
>
> b. Indeed, when training simulation-based neural SDEs, small integration steps are usually adopted to reduce discretization errors. However, our method does not require integration during training.
>
> During inference (testing), simulation-based neural SDE methods typically use small steps during integration to reduce discretization errors. Likewise, our method can also use small steps during integration for reducing discretization errors; there is no difference here.
>
> Neither method can eliminate statistical errors caused by the dataset sampling process itself. It’s crucial to distinguish between discretization errors from integration approximations and statistical errors from data sampling. From an information-theoretic perspective, even simulation-based methods cannot accurately recover lost signals when sampling intervals are large; they merely learn one possible smoothed version. Our linear interpolation assumption also yields another possible smoothed result, and we cannot judge which is superior because the ground truth is unknown. A more extreme example is the rectified flow's "reflow" process (see Algorithm 1, Figure 3 in https://arxiv.org/pdf/2209.03003), where curves induced by linear assumptions are straightened into lines. Neither our method nor simulation methods exacerbate or reduce statistical errors caused by sampling intervals. Furthermore, The statistical errors inherent in the data often overshadow the differences among various assumed dynamics—whether the dynamics are derived from numerical integration or predefined as linear. A classic example is the Kalman filter. Although linear dynamics may not always represent the optimal model, the impact of such modeling assumptions becomes negligible compared to the dominant statistical noise(e.g. https://ieeexplore.ieee.org/document/1243440). What ultimately matters is usability and robustness, which is precisely why the Kalman filter has been so successful in practice(https://ieeexplore.ieee.org/document/6400245).
>
> We've also tried other interpolation methods (e.g., cubic Hermite) on pusht and video tasks. Compared to linear interpolation, the performance is sometimes slightly better, sometimes slightly worse, without significant differences. Additionally, related works (e.g., https://arxiv.org/pdf/2410.22918, section "Nonlinear Predefined Dynamics") discuss predefined dynamics, indicating that linear dynamics are often sufficient in practice.

---

> > ### Author Response · Authors · 2025-08-06
> >
> > ## Proof of Consistency for the Decoupled SDE Estimators
> >
> > We aim to prove that the two-stage estimation procedure for the drift term
> > $\hat{f}$ and the diffusion term $\hat{g}$ is statistically consistent. This means that in the limit of infinite data and with sufficient model capacity, the learned functions converge in probability to the true underlying functions ($\hat{f} \to f_{\text{true}}, \hat{g} \to g_{\text{true}}$).
> >
> > The true data-generating process is given by the SDE:
> >
> > $$
> > dX_t = f_{\text{true}}(X_t) dt + g_{\text{true}}(X_t) dW_t
> > $$
> >
> > From the Euler-Maruyama discretization, the target variable for our estimation, $Y = \frac{\Delta X}{\Delta t}$, can be written as:
> >
> > $$
> > Y = f_{\text{true}}(X_t) + g_{\text{true}}(X_t) \cdot \frac{\Delta W}{\Delta t}
> > $$
> >
> > The term $Z = g_{\text{true}}(X_t) \cdot \frac{\Delta W}{\Delta t}$ is a zero-mean noise term with a symmetric distribution (specifically, Gaussian).
> >
> > ---
> >
> > ## Part 1: Consistency of the Flow Estimator ($\hat{f}$)
> >
> > The estimator $\hat{f}$ is found by minimizing the expected desingularized logarithmic loss:
> >
> > $$
> > \hat{f} = \arg\min \mathbb{E} \left[ \log\left( (\hat{f}(X) - Y)^2 + \delta \right) \right]
> > $$
> >
> > where $\delta > 0$ is the desingularization constant. To find the function $\hat{f}$ that minimizes this expectation for a given $X$, we take the functional derivative with respect to $\hat{f}(X)$ and set it to zero:
> >
> > $$
> > \frac{\partial}{\partial \hat{f}} \mathbb{E}\left[\log\left((\hat{f} - Y)^2 + \delta\right)  \big|  X \right] = \mathbb{E}\left[ \frac{2(\hat{f} - Y)}{(\hat{f} - Y)^2 + \delta}  \big|  X \right] = 0
> > $$
> >
> > We must show that the true function, $\hat{f}(X) = f_{\text{true}}(X)$, satisfies this condition. Substituting $\hat{f} = f_{\text{true}}$ and $Y = f_{\text{true}} + Z$:
> >
> > $$
> > \mathbb{E}\left[ \frac{2(f_{\text{true}} - (f_{\text{true}} + Z))}{(f_{\text{true}} - (f_{\text{true}} + Z))^2 + \delta}  \big|  X \right] = \mathbb{E}\left[ \frac{-2Z}{Z^2 + \delta}  \big|  X \right]
> > $$
> >
> > The function $h(Z) = \frac{-2Z}{Z^2 + \delta}$ is an odd function, meaning $h(-Z) = -h(Z)$. The expectation is taken over the noise distribution of $Z$, which is symmetric around zero. The expected value of an odd function with respect to a zero-mean symmetric probability distribution is zero.
> >
> > Thus, the true drift $f_{\text{true}}$ is the unique minimizer of the expected loss. By the properties of M-estimators, this implies that the estimator $\hat{f}$ is consistent.
> >
> > ---
> >
> > ## Part 2: Consistency of the Diffusion Estimator ($\hat{g}$)
> >
> > The estimator for the diffusion variance, $\hat{g}^2$, is found by minimizing the residual matching loss, which is a standard Mean Squared Error (MSE) loss:
> >
> > $$
> > \hat{g}^2 = \arg\min \mathbb{E} \left[ \left( \hat{g}^2(X) - (\hat{f}(X) - Y)^2 \Delta t \right)^2 \right]
> > $$
> >
> > The value of $\hat{g}^2(X)$ that minimizes this expected squared error is the conditional expectation of its target:
> >
> > $$
> > \hat{g}^{2\ast}(X) = \mathbb{E} \left[ (\hat{f}(X) - Y)^2\Delta t  \big|  X \right]
> > $$
> >
> > The consistency of this stage relies on the consistency of the first stage.
> >
> > From Part 1, we know that as the amount of data approaches infinity, $\hat{f}(X) \to f_{\text{true}}(X)$.
> >
> > Therefore, the target for the diffusion estimator converges to the true conditional variance of the residual:
> >
> > $$
> > \hat{g}^{2\ast}(X) \to \mathbb{E} \left[ (f_{\text{true}}(X) - Y)^2 \Delta t  \big|  X \right]
> > $$
> >
> > We substitute $Y = f_{\text{true}}(X) + Z = f_{\text{true}}(X) + g_{\text{true}}(X)  \frac{\Delta W}{\Delta t}$:
> >
> > $$
> > \mathbb{E} \left[ (-g_{\text{true}}(X)  \frac{\Delta W}{\Delta t})^2\Delta t  \big|  X \right]
> > = \mathbb{E} \left[ g_{\text{true}}^2(X)  \frac{(\Delta W)^2}{(\Delta t)^2}  \Delta t  \big|  X \right]
> > $$
> >
> > Since $\Delta W \sim \mathcal{N}(0, I \Delta t)$, the expectation of the squared noise increment is its variance, i.e., $\mathbb{E}[(\Delta W_i)^2] = \Delta t$.
> >
> > Hence,
> >
> > $$
> > \mathbb{E} \left[ g^2_{\text{true}, i} \frac{(\Delta W_i)^2}{(\Delta t)^2} \Delta t \right]
> > = g^2_{\text{true}, i}  \frac{\mathbb{E}[(\Delta W_i)^2]}{(\Delta t)^2}  \Delta t
> > = g^2_{\text{true}, i}  \frac{\Delta t}{(\Delta t)^2}  \Delta t
> > = g^2_{\text{true}, i}
> > $$
> >
> > The optimal estimator for the diffusion variance converges to the true diffusion variance, $g_{\text{true}}^2(X)$. Therefore, the estimator $\hat{g}$ is also consistent.

---

> > > ### Comment · Reviewer_yZgJ · 2025-08-07
> > >
> > > Thank you to the authors for providing the detailed derivation. I found the result in Part I particularly interesting—it nicely leverages the symmetry of the noise distribution.
> > >
> > > Just a quick question: have the authors compared the proposed method with existing approaches based on Euler–Maruyama discretization, including moment-matching-based techniques? Such a comparison would help contextualize the advantages or differences of the proposed approach.
> > >
> > > In addition, the reply on the linear interpolation seems incomplete.

---

> ### Author Response · Authors · 2025-08-08
>
> We sincerely thank you for your continued engagement. We have completed the discussion on the linear interpolation part in the last comment. We have investigated the moment-matching method and even tried to implement the HBR method described in https://journals.aps.org/pre/pdf/10.1103/PhysRevE.106.014140 on the PushT dataset.
> The estimation function in HBR is
>
> $f(x_i) = \frac{1}{\tau}
> \frac{\sum_{k=1}^N \mathbf{1}\left( X(t_k) \in B_x(x_i) \right) \, \Delta X_k}
> {\sum_{k=1}^N \mathbf{1}\left( X(t_k) \in B_x(x_i) \right)}
> $
>
> $G(x_i)G(x_i)^\top = \Sigma(x_i) = \frac{1}{\tau}
> \frac{\sum_{k=1}^N \mathbf{1}\left( X(t_k) \in B_x(x_i) \right)  \Delta X_k  \Delta X_k^\top}
> {\sum_{k=1}^N \mathbf{1}\left( X(t_k) \in B_x(x_i) \right)}$
>
> if $G(x_i)$ is diagonal:
>
> $G_{\ell \ell}(x_i) = \sqrt{\left[ \Sigma(x_i) \right]_{\ell\ell}} , \quad \ell = 1, \dots, d.$
>
> However, we found it different from our method. Consider a uniform linear motion:
> according to the estimate of $G$, we have
>
> $G = \sqrt{\frac{1}{\tau}  \mathbb{E}[(v \tau)^2]} = v \sqrt{\tau}$
>
> which is nonzero for finite $\tau$, but tends to zero as $\tau \to 0$. This means that such a moment-matching method requires extremely high data density (as is indeed the case in the experiments shown in the paper you provided).
> In tasks with even slightly higher dimensionality, this is practically impossible. Moreover, this method requires binning; in a high-dimensional space, the data become extremely sparse and are concentrated near the boundaries, leaving most hypercubes empty. Even if it is one-dimensional, the method can be complex when the sampling rate is low (https://journals.aps.org/pre/abstract/10.1103/PhysRevE.83.066701). Therefore, this method is not applicable to our task.

---

> > ### Comment · Reviewer_yZgJ · 2025-08-08
> >
> > Thank you to the authors for the very thorough discussion. I now agree that the paper indeed arrives at some interesting conclusions, and I would be willing to raise my rating for the paper. However, regarding the linear interpolation component, I would still like to point out that while flow matching and diffusion models can indeed use interpolation-like approaches to obtain meaningful dynamical models, the resulting dynamical systems are generally time-inhomogeneous. This is different from the time-homogeneous SDE model considered in this paper (Equation (9)). Therefore, if simple linear interpolation can truly yield a valid estimate of a time-homogeneous SDE, the rationale for its validity cannot be directly borrowed from the arguments used in the context of flow matching or related methods.

---

> > > ### Author Response · Authors · 2025-08-09
> > >
> > > Thank you for your excellent question on time-homogeneous and time-inhomogeneous. We would like to share the debate and thinking process among authors. We have thought about this problem from the very beginning. First, I must point out that if we start from a time-inhomogeneous SDE, our method and loss are still valid, we just need to insert t into the drift and diffusion terms. In that case, arguments from flow-matching and stochastic interpolants can be applied to our method and the task's performance will be a bit better (according to my memory, about a 10% FVD decrease on the video task). So why did we later remove t? For generative models, before the neural network learns the generation process, this dynamic process, one cannot assume it is time-homogeneous, because without knowing if the underlying process is time-homogeneous, this is a very strong inductive bias, so they need to start from a time-inhomogeneous one. But if we already knew that the process to be learned is time-invariant, then directly putting this inductive bias into the model assumption could possibly be a good choice, because we expect this will force the model to learn a time-invariant process, thereby getting closer to the real physical process. Furthermore, a time-invariant process implies time through the state.  It will be really appreciated if you raise your score to be equal to or greater than 4.

---

### Official Review · Reviewer_Rk5z · 2025-07-09

**Clarity:** 3
**Significance:** 2
**Originality:** 2
**Rating:** 4
**Confidence:** 3

**Summary:**

The paper argues that sequence-modelling problems like time-series are better viewed as samples from an underlying continuous-time dynamical system. Instead of modelling them as bridge from noise to data paradigm like in diffusion or flow-matching, this paper proposes to learn a Neural Stochastic Differential Equation (SDE) with both the flow (drift) and diffusion terms, parameterized by neural networks and trained by a maximum-likelihood objective that doesn't require simulation for gradient estimation. Neural SDE gives comparable benchmark results and exhibits power-law scaling.

**Questions:**

1) How hard is it to learn a low-rank instead of diagonal diffusion covariance? Do you have any early results?

2) Have you tried augmenting the state with learned memory or sliding window to relax the time invariant Markov assumption?

3) Can you show scaling experiments on some other datasets like Something-Something?

**Ethical Concerns:**

["NO or VERY MINOR ethics concerns only"]

**Limitations:**

yes

**Quality:**

3

**Strengths And Weaknesses:**

Strengths
1) Their simulation free maximum-likelihood method with a decoupled two-stage optimizer avoids the costly forward SDE unrolling required in traditional gradient estimation.

2) They model both drift and diffusion as learning the parameters of an SDE that describes the evolution of continuous sequences, which eliminates the need for discretization or high transport costs associated with iterative methods.

3) It shows inference efficiency due to directly modelling continuous-time dynamics—rather than discretely transitioning from Gaussian noise as in diffusion and flow matching approaches.


Weaknesses
1) Their time-independent formulation depends only on 𝑋𝑡, ignoring long-range context. It has impact on tasks where history matters for e.g. on KTH, visually similar clips generated unintended action changes.

2) They model diffusion coefficients with diagonal covariance. This independent-noise assumption may under-fit correlated high-dimensional systems.

3) They also need to tune the weighting coefficient, α in the denoiser network manually per dataset, which can be cumbersome.

4) Scaling curve shown only for CLEVRER dataset. So it is unclear how the proposed method will perform on realistic large-scale video dataset (e.g., Something-Something[1])

[1] Goyal, Raghav, et al. "The" something something" video database for learning and evaluating visual common sense." Proceedings of the IEEE international conference on computer vision. 2017.

---

> ### Author Rebuttal · Authors · 2025-07-31
>
> We sincerely thank you for the positive assessment. The weaknesses and future directions you have pointed out—namely, relaxing the Markov assumption, exploring richer diffusion covariances, and scaling to larger datasets—are indeed the most critical next steps. These align perfectly with the limitations we discussed in Section 6 of our paper, and we appreciate the opportunity to elaborate on our perspective regarding these important questions.
>
> On Learning a Low-Rank vs. Diagonal Diffusion Covariance
> Your question about learning a richer covariance structure is a critical one for advancing this line of research. The primary reason for choosing a diagonal covariance was to ensure computational tractability, a choice also made in standard Latent SDE implementations for memory efficiency (https://arxiv.org/pdf/2001.01328 section 3.3). While this diagonal assumption limits flexibility, our method does not introduce restrictions beyond those already common in the field.
>
> Learning a low-rank approximation is an exciting possibility. However, it would pose a significant challenge to the decoupled optimization scheme we derived, which relies on the simplicity of the diagonal structure. We have not yet attempted a low-rank approximation, but we see it as a valuable, albeit non-trivial, next step for future work.
>
> On Relaxing the Time-Invariant Markov Assumption
> This is an excellent suggestion. As we acknowledge in our limitations section, the Markov assumption is a key simplification. While we have not implemented a full memory mechanism, we believe that techniques from the video diffusion community could be adapted here. For example, one could condition the SDE on a history vector, $H_t$, similar to how classifier-free guidance works. This history could be encoded by a recurrent or transformer-based network and fed into the drift and diffusion functions, explicitly allowing the model to capture longer-range dependencies. We are hopeful that our foundational work will inspire future explorations in this direction.
>
> On Scaling Experiments with Larger Datasets
> We agree that testing our model's scaling properties on larger and more complex datasets, such as Something-Something, would be a fantastic validation of its potential. Due to the significant computational resources required for such scaling experiments, we have not yet been able to run these tests. Should our work be accepted by the community, demonstrating its performance on larger-scale benchmarks would be a top priority for us.

---

> > ### Comment · Area_Chair_RdC4 · 2025-08-04
> >
> > Dear Reviewer,
> >
> > Please respond to the rebuttal. Thanks.
> >
> > AC.

---

> > > ### Comment · Area_Chair_RdC4 · 2025-08-05
> > >
> > > Dear Reviewer,
> > >
> > > Please comment on the rebuttal. Thanks.
> > >
> > > AC.

---

### Decision · Program_Chairs · 2025-09-17

**Decision:**

Reject

**Comment:**

This submission proposes a simulation-free training method for Neural SDEs applied to sequence modeling using decoupled drift-diffusion optimization. While some reviewers acknowledged that their initial concerns were partially addressed through author rebuttals, Reviewer oK7p remained negative and found the work confusing. No reviewer provided strong advocacy for acceptance—even those who increased their scores maintained only borderline positions. Reviewer Rk5z stated that "several of my concerns were not addressed due to computational and time limitations. While the work is interesting, it would benefit from engaging more directly with these questions." Reviewer yZgJ stated "the work would benefit from a stronger focus on SDE modeling, along with more targeted experimental design and evaluation, which would also enhance the theoretical clarity of the paper." Overall, the work still falls below the threshold of acceptance.